# PRETRAINING SCALING LAWS FOR GENERATIVE EVALUATIONS OF LANGUAGE MODELS

**Rylan Schaeffer**[*]
Computer Science
Stanford University

**Noam Levi**[*]
AI Center
EPFL

**Brando Miranda**
Computer Science
Stanford University

**Sanmi Koyejo**
Computer Science
Stanford University

## ABSTRACT

Neural scaling laws have driven the field's ever-expanding exponential growth in parameters, data and compute. While scaling behaviors for pretraining losses and discriminative benchmarks are well established, generative benchmarks such as mathematical problem-solving or software engineering remain under-explored. We propose and evaluate three different pretraining scaling laws for fitting pass-at-$k$ on generative evaluations and for predicting pass-at-$k$ of the most expensive model using cheaper models. Our three scaling laws differ in the covariates used: (1) pretraining compute, (2) model parameters and pretraining tokens, (3) log likelihoods of gold reference solutions. First, we demonstrate that generative evaluations introduce new hyperparameters (in our setting, $k$) that act as a control lever for scaling behavior, modulating both the scaling law parameters and the predictability of performance. Second, we identify a stark difference in parameter stability: while the compute and parameters+tokens laws stabilize for only the last $\sim 1.5-2.5$ orders of magnitude, the gold reference likelihood law is uniquely stable, converging across $\sim 5$ orders. Third, in terms of predictive performance, we find all three scaling laws perform comparably, although the compute law predicts slightly worse for small $k$ and the gold reference law predicts slightly worse for large $k$. Finally, we establish a theoretical connection, proving that the compute scaling law emerges as the compute-optimal envelope of the parameters-and-tokens law. Our framework provides researchers and practitioners with insights and methodologies to forecast generative performance, accelerating progress toward models that can reason, solve, and create.

## 1 INTRODUCTION

Neural scaling laws, which predictably map resources to the performance of neural networks, are foundational for developing frontier AI systems (Kaplan et al., 2020b; Hoffmann et al., 2022b). Such empirical regularities for improving model performance from scaling parameters, data, and compute have driven pursuit of ever-larger models. Much of the research in neural scaling laws has focused on fitting and predicting model performance on pretraining losses, e.g., (Kaplan et al., 2020b; Hoffmann et al., 2022b; Clark et al., 2022; Hernandez et al., 2022; Sardana et al., 2024; Muennighoff et al., 2023; Porian et al., 2024; Pearce & Song, 2024) and on "downstream" *discriminative* evaluations like multiple-choice question-answering, e.g., (Schaeffer et al., 2023; Gadre et al., 2024; Schaeffer et al., 2024b; Grattafiori et al., 2024; Chen et al., 2025; Bhagia et al., 2025).

While pretraining losses and discriminative evaluations are undeniably useful, many capabilities we care about are *generative*: writing books, autoformalizing mathematics, or conducting cutting-edge research. Scaling laws for generative tasks remain undercharacterized (Appendix A) (but see

---

[*]Denotes equal authorship. Contact: rschaef@cs.stanford.edu, noam.levi@epfl.ch, sanmi@cs.stanford.edu.

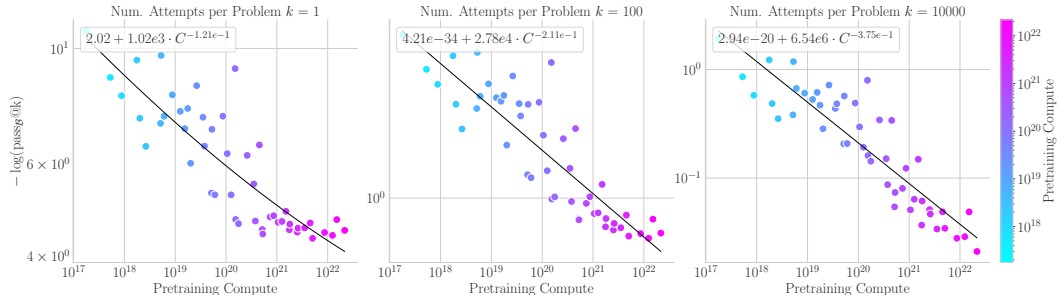

Figure 1: **Scaling of GSM8K Pass Rates with Pretraining Compute (Full Fit).** Each panel fits Eqn. 3 $-\log\big(\text{pass}_{\mathcal{B}}@k\big)(C, k) = E_0(k) + C_0(k) \cdot C^{-\alpha(k)}$ to GSM8K pass rates for Pythia checkpoints across $\sim$5 orders of magnitude of pretraining compute. Increasing $k$ drives two major shifts: (i) The irreducible error $E_0(k)$ vanishes (from $\approx 2 \to 0$), removing the performance plateau, and (ii) the power law steepens, with the exponent $\alpha(k)$ rising from $\approx 0.121 \to 0.375$. **Takeaway:** Larger $k$ eliminate irreducible error and yield steeper pass-at-$k$ rates with respect to pretraining compute.

(OpenAI et al., 2024a; Hu et al., 2024)); for example, Gadre et al. (2024) studied 46 tasks, *none* of which were generative. Generative evaluations differ from discriminative evaluations in many ways, most crucially that performance is calculated from the model's open-ended samples, which introduces new considerations such as how attempts are sampled from the model, including the sampling temperature and the sampling algorithm, and which metric is used to measure performance.

In this work, we propose a framework for fitting scaling laws and predicting performance on generative evaluations, focusing on a particular setting: benchmarks with verifiable binary rewards, with multiple attempts per problem, with performance scored using the "pass-at-$k$" metric (Kulal et al., 2019; Chen et al., 2021). We chose this setting due to its widespread use in the literature, e.g., (First et al., 2023; Brown et al., 2024; Hassid et al., 2024; Hughes et al., 2024; Chen et al., 2024; Ehrlich et al., 2025; Kwok et al., 2025). In this setting, we make the following contributions:

1. We propose three scaling laws that fit and predict pass-at-$k$ from three different covariates: (1) pretraining compute (Sec. 3), (2) model parameters and pretraining tokens (Sec. 4) and (3) likelihoods of gold reference solutions (Sec. 5).

2. We reveal that a hyperparameter specific to this setting – the number of attempts per problem $k$ – offers a new lever to shape the scaling and predictability of model performance.

3. We quantify the stability of each scaling law's parameters. We find that parameters for the compute scaling law and the parameters + tokens scaling law stabilize by the last $\sim$1.5$-$2.5 orders of magnitude of pretraining compute, whereas parameters for the gold reference likelihood scaling law are stable over the last $\sim$5 orders of magnitude.

4. We quantify how predictive each scaling law is, finding that all three comparably predict the performance of the most expensive model based on cheaper models. As a lesser difference, we find that the compute scaling law has slightly higher predictive error for small $k$, whereas the gold reference likelihoods law has slightly higher predictive error for large $k$.

5. We prove that the compute-only scaling law is the compute-optimal envelope of the parameters-and-tokens scaling law, obtained by minimizing the objective under a fixed compute budget. From this, we derive a dimensionless *misallocation penalty* that explains quantitatively when and why pretraining recipes underperform compute-optimal scaling.

## 2 METHODOLOGY

**Language Model Family:** For our experiments, we used the Pythia family (Biderman et al., 2023) of 8 models from 14M to 12B parameters pretrained on up to 300B tokens from The Pile (Gao et al., 2020). We restricted our analysis to Pythia family because it is, to our knowledge, the only public model family meeting three criteria: (1) densely sampled across both model parameters ($N$) and pretraining tokens ($D$); (2) spanning many orders of magnitude in pretraining compute (5); and (3)

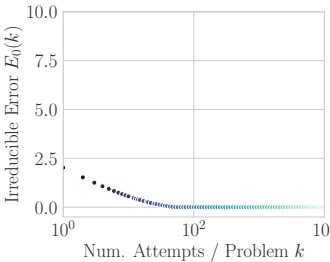 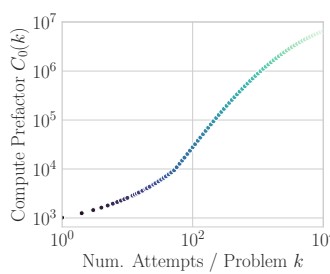 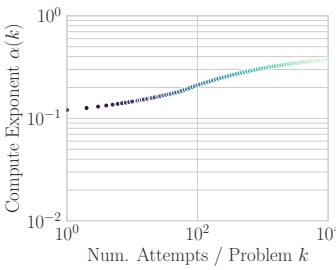

Figure 2: **Number of Attempts per Problem $k$ Shapes Scaling Law Parameters (Full Fit).** We fit Eqn. 3 to each $k$ (hue); see Appendix D for the next two scaling laws. **Left:** Irreducible error $E_0(k)$ decays roughly exponentially with $k$ and is $\approx 0$ by $k \approx 10^2$. **Center:** Compute prefactor $C_0(k)$ increases monotonically with $k$, indicating that once $E_0(k) \to 0$, the compute-dependent term dominates. **Right:** Compute exponent $\alpha(k)$ increases smoothly, from $\sim 0.15$ at $k{=}1$ to $\sim 0.3$ at $k{=}10^4$, indicating that larger sampling budgets yield steeper, more favorable scaling behaviors.

known token budgets per checkpoint. Other families (e.g., Llama 3, Qwen 3) either lack sufficient intermediate checkpoints to robustly fit a 5-parameter scaling law or do not disclose specific per-model pretraining token budgets. We used 8 checkpoints per parameter size of the non-deduplicated variants, but excluded extremely overtrained checkpoints identified as abnormal (Godey et al., 2024). We approximated pretraining compute as $C \approx 6\,N\,D$ (Kaplan et al., 2020b; Hoffmann et al., 2022b; Pearce & Song, 2024; Gadre et al., 2024; Schaeffer et al., 2024b; Porian et al., 2024) and used intermediate checkpoints in lieu of fully trained models.

**Benchmarks:** We used two well-established generative tasks: **GSM8K** (Cobbe et al., 2021) with $8,500$ grade school math word problems, and **MATH** (Hendrycks et al., 2021) with $12,500$ competition mathematics problems. Each attempt at solving a problem yields a binary outcome: success or failure. Because generative evaluations are computationally expensive, to enable a rigorous analysis across 5 orders of magnitude of compute with high sample counts, we evaluated all checkpoints on a randomly selected subset of 128 problems from GSM8K and 128 problems from MATH; see Appendix B for the number of samples per problem per model, and Appendix C for the costs.

**Metric:** We measured the performance of each model using the pass-at-$k$ metric (Kulal et al., 2019), defined as the probability that if a model makes $k$ attempts on the $i$th problem, at least one attempt is successful. We estimate this per-problem pass rate using the unbiased, low variance and numerically stable estimator of Chen et al. (2021), which, for each problem, draws $n_i > k$ samples, counts the number of successes $s_i \in [0, n_i]$, and then averages over all subsets of size $k$:

$$\text{pass}_i@k \quad \stackrel{\text{def}}{=} \quad 1 - \frac{\binom{n_i - s_i}{k}}{\binom{n_i}{k}} \tag{1}$$

For a benchmark $\mathcal{B}$ of $|\mathcal{B}|$ problems, a model's benchmark pass rate at $k$ attempts is estimated as:

$$\text{pass}_\mathcal{B}@k \quad \stackrel{\text{def}}{=} \quad \frac{1}{|\mathcal{B}|} \sum_{i \in \mathcal{B}} \text{pass}_i@k \tag{2}$$

Pass rates are sometimes phrased as the fraction of problems in the benchmark solved ("coverage"), which is mathematically equivalent (Brown et al., 2024; Schaeffer et al., 2025b). Importantly, pass-at-$k$ is fundamentally different from (discriminative) accuracy, as pass-at-$k$ is a continuous probability derived from the model's generative distribution whereas accuracy is a discriminative function of the correct and incorrect multiple choices (Appendix A).

**Generation:** Generative evaluation introduces a vast hyperparameter space (e.g., temperature (Ackley et al., 1985), top-p (Holtzman et al., 2020), top-k (Fan et al., 2018)). We deliberately isolated the number of attempts per problem ($k$) as our primary variable because it is a simple yet effective lever for scaling inference-time compute (OpenAI et al., 2024b; Snell et al., 2024; Brown et al., 2024; DeepMind et al., 2025; DeepSeek-AI et al., 2025a) and because previous work has studied the effect of sampling algorithm and sampling temperature on GSM8K (Chen et al., 2021; Schaeffer et al., 2025a). We used temperature-only sampling at $\tau = 1.0$, holding the decoding algorithm fixed.

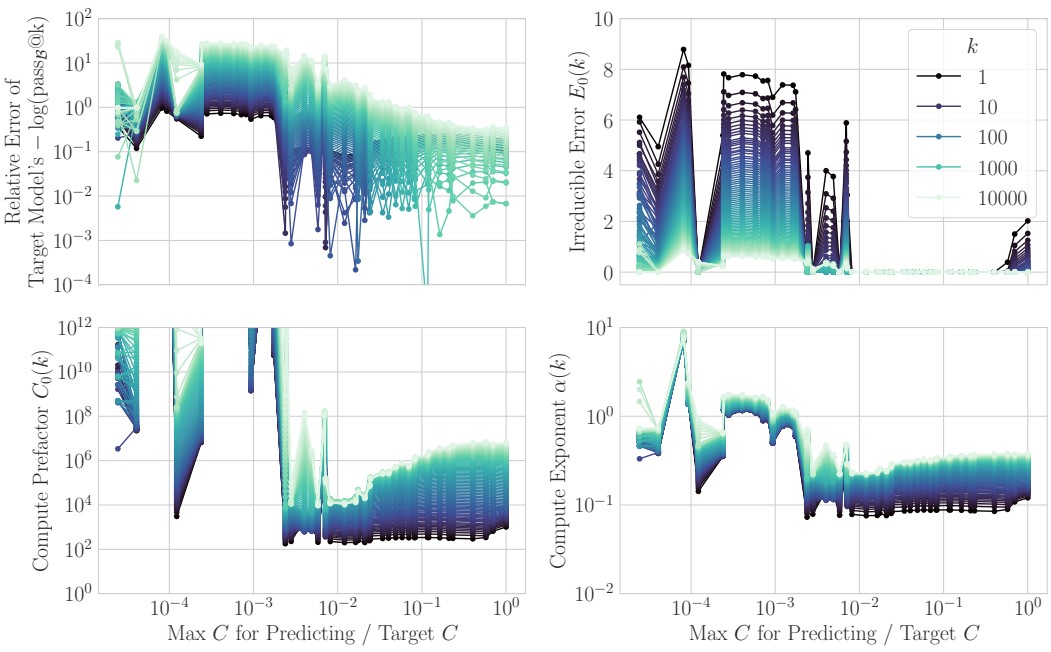

Figure 3: **Predicting GSM8K Pass Rates from Scaling Pretraining Compute (Backtesting).** We evaluate predictability via *backtesting*: iteratively fitting Eq. 3 on subsets of models ($C \leq C_{\max}$) to predict the most expensive model (Pythia 12B-parameter 300B-token; $\approx 2.16 \times 10^{22}$ FLOP)'s $-\log(\text{pass}_\mathcal{B}@k)$. The $x$-axis denotes the compute horizon relative to the target ($C_{\max}/C_{\text{target}}$). **Top Left:** Relative error decreases and then plateaus: reliable prediction requires checkpoints within $\sim 2$ orders of magnitude of the target for $k \in \{1, 10^2\}$ and $\sim 1.5$ for $k = 10^4$. **Other Three Panels:** Backtested estimates of the scaling law parameters are initially unstable but converge to their full-fit values once fits include models within $\sim 2$ orders of magnitude of the target.

## 3  FITTING AND PREDICTING PASS RATES FROM PRETRAINING COMPUTE

For our first scaling law, we assessed how well the benchmark $\text{pass}_\mathcal{B}@k$ can be (1) fit as a function of pretraining compute $C$ and (2) predicted from pretraining compute.

### 3.1  FITTING PASS RATES FROM PRETRAINING COMPUTE

Motivated by the GPT-4 Technical Report (OpenAI et al., 2024a), we posited that the negative log of benchmark pass rates follows a scaling law as a function of model pretraining compute $C$:

$$-\log\left(\text{pass}_\mathcal{B}@k\right)(C, k) = E_0(k) + \frac{C_0(k)}{C^{\alpha(k)}}, \tag{3}$$

where $E_0(k) \geq 0$ is the irreducible error, $C_0(k) > 0$ is the compute scaling prefactor, and $\alpha(k) > 0$ is the compute scaling exponent. In contrast with prior research on scaling laws, parameters that were previously constant are now parameterized by the number of attempts per problem $k$. Thus, $k$ can be viewed as a hyperparameter that offers a new lever to change the predictability of scaling laws in a way that is not available for pretraining loss or discriminative evaluation scaling laws.

We fit the scaling law parameters based on the model checkpoints' pretraining compute and their corresponding $\text{pass}_\mathcal{B}@k$. Visually, the fit scaling laws captured the trend reasonably well for different values of number of attempts per problem $k$ (Fig. 1). We next evaluated what role the number of attempts per problem $k$ has on the scaling law parameters: As $k$ increases, the irreducible error term $E_0(k)$ falls roughly exponentially with $k$ and is effectively 0 by $k \approx 1 \times 10^2$ (Fig. 2 Left). This is consistent with the probability a problem goes unsolved falling exponentially with the number of attempts per problem (Levi, 2025; Schaeffer et al., 2025b). Thus, by increasing $k$, Eqn. 3 becomes a pure power law with no irreducible error. In comparison, as $k$ increases, the compute prefactor and

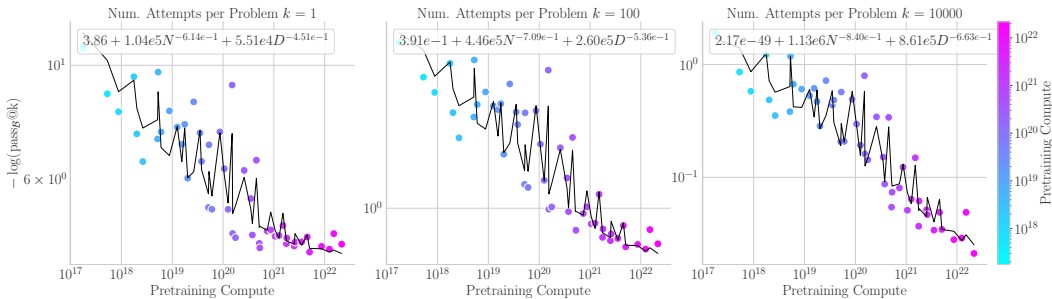

Figure 4: **Scaling of GSM8K Pass Rates with Parameters and Tokens (Full Fit).** Each panel fits Eqn. 4, $-\log\big(\mathrm{pass}_\mathcal{B}@k\big)(N, D, k) = \mathcal{E}_0(k) + N_0(k)\, N^{-\beta(k)} + D_0(k)\, D^{-\gamma(k)}$. Decomposing compute into parameters $N$ and tokens $D$ instead yields tighter in-range fits than the compute law for all $k$. Consistent with Fig. 2, the irreducible error decreases sharply with $k$ ($\mathcal{E}_0: 3.87 \to 0$ by $k \approx 300$), after which variation is dominated by $(N, D)$ terms. However, despite the better global fit, the largest-compute model checkpoint in each panel exhibits comparatively large relative error.

compute exponent also increase, with the compute prefactor rising $\sim$4 orders of magnitude (Fig. 2 Center) and the compute exponent rising moderately from $1.21 \times 10^{-1}$ to $3.75 \times 10^{-1}$ (Fig. 2 Right).

## 3.2 PREDICTING PASS RATES FROM PRETRAINING COMPUTE

Previously, we fit Eqn. 3 using all available checkpoints. However, a key motivation for scaling laws is *prediction*: can we forecast the performance of an expensive model using cheaper models?

We evaluated the predictability of pass rates via backtesting, a cross-validation-like process for evaluating forecasts in which cheaper models are used to predict the performance of the most expensive model. We fixed a target model checkpoint pretrained on $C_{\mathrm{target}}$ FLOP; in our work, we used Pythia-12B trained for 300B tokens, pretrained with approximately $2.16 \times 10^{22}$ FLOP. For a sequence of compute caps $C_{\mathrm{max}} \le C_{\mathrm{target}}$, we: (i) fit Eq. 3 on all model checkpoints with compute $C \le C_{\mathrm{max}}$ to obtain $\widehat{E}_0(k)$, $\widehat{C}_0(k)$, and $\widehat{\alpha}(k)$; (ii) extrapolate to the target compute horizon, and (iii) measure the absolute relative error to the target model's $-\log\big(\mathrm{pass}_\mathcal{B}@k_{\mathrm{target}}\big)$:

$$\mathrm{RelativeError}(k, C_{\mathrm{max}}) \overset{\mathrm{def}}{=} \frac{\mathrm{abs}|-\log\big(\mathrm{pass}_\mathcal{B}@k_{\mathrm{target}}\big) - \widehat{E}_0(k) - \widehat{C}_0(k) \cdot C_{\mathrm{target}}^{-\widehat{\alpha}(k)}|}{-\log\big(\mathrm{pass}_\mathcal{B}@k_{\mathrm{target}}\big)}.$$

We report results as a function of the compute ratio $C_{\mathrm{max}}/C_{\mathrm{target}}$.

Relative errors typically decrease and then plateau as higher-compute checkpoints are included in the fit (Fig. 3, top left). For $k \in \{1, 1e2\}$, reliable extrapolation requires that the largest checkpoint used for fitting is within $\sim$2 orders of magnitude of the target's compute, and the relative errors plateau at $1 \times 10^{-1}$. In comparison, for $k = 1e4$, reliable extrapolation requires $\sim$1.5 order of magnitude, and the relative errors plateau below 1. Similarly, the backtested estimates of the parameters $\widehat{E}_0(k), \widehat{C}_0(k)$ and $\widehat{\alpha}(k)$ converge to their full-fit values once models within $\sim$1.5–2.5 orders of magnitude of $C_{\mathrm{target}}$ are included (Fig. 3, other three panels). For these models on this benchmark, the compute scaling law provides useful forecasts so long as forecasts are made using models within $\sim$2 orders of magnitude of compute as the target model; when only smaller checkpoints are available, the predictive error increases and the fitted parameters are unreliable.

## 4 FITTING AND PREDICTING PASS RATES FROM PARAMETERS AND TOKENS

In our second approach, we assessed how well the benchmark $\mathrm{pass}_\mathcal{B}@$k can be (1) fit and (2) predicted as a function of the number of model parameters $N$ and pretraining tokens $D$.

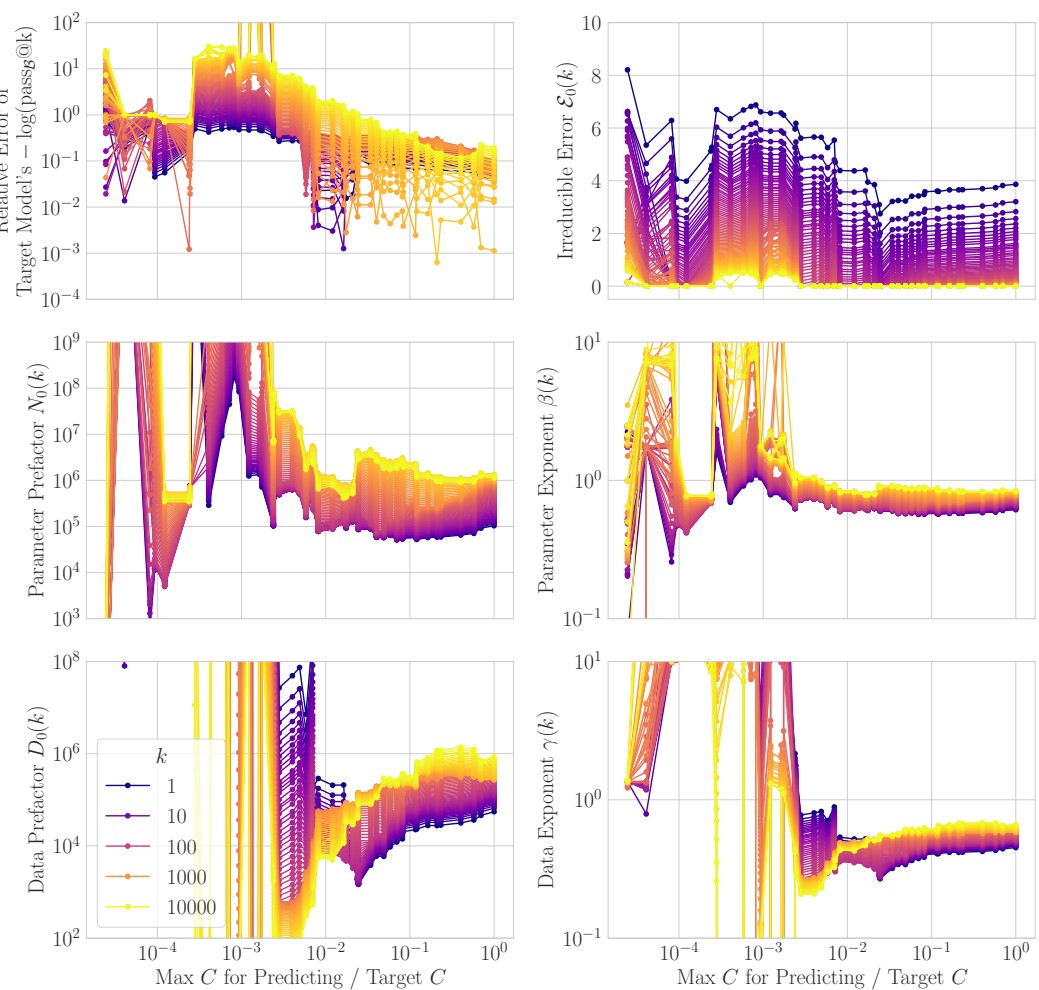

Figure 5: **Predicting GSM8K Pass Rates from Scaling Parameters and Tokens (Backtesting).**
We evaluate how accurately the parameters + tokens scaling law (Eqn. 4) predicts the most expensive
model's $-\log(\text{pass}_\mathcal{B}@k)$. **Top Left:** The relative error decreases as higher-compute checkpoints
are included, plateauing once the fit includes models within $\sim 2$ orders of magnitude of the target.
**Other Five Panels:** Estimates of the five scaling law parameters are initially unstable but converge
to their full-fit values once included models are within $\sim 2$ orders of magnitude of the target.

## 4.1 FITTING PASS RATES FROM MODEL PARAMETERS AND PRETRAINING TOKENS

Motivated by Hoffmann et al. (2022b), we posited that the negative log of benchmark pass rates
might instead follow a scaling law as a function of model parameters $N$ and pretraining tokens $D$:

$$-\log\big(\text{pass}_\mathcal{B}@k\big)(N, D, k) = \mathcal{E}_0(k) + \frac{N_0(k)}{N^{\beta(k)}} + \frac{D_0(k)}{D^{\gamma(k)}}, \qquad (4)$$

where $\mathcal{E}_0(k)$ is the irreducible error, $N_0(k)$ is the parameter scaling prefactor, $\beta(k)$ is the parameter
scaling coefficient, $D_0(k)$ is the token scaling prefactor and $\gamma(k)$ is the token scaling coefficient.

We fit this parameters + tokens scaling law. Visually, this law provides a better characterization
of performance for the full range of models, with the fitted curves appearing tighter and reducing
the residual scatter (Fig.4), compared against the pretraining compute law (Fig.1). Consistent with
our findings for the compute-only law, the irreducible error term $\widehat{\mathcal{E}}_0(k)$ decreases with the number
of attempts per problem $k$, although it does so more gradually, reaching zero by $k \approx 3 \times 10^2$
(Fig. 12). However, despite the improved global fit, we observed that the largest-compute model

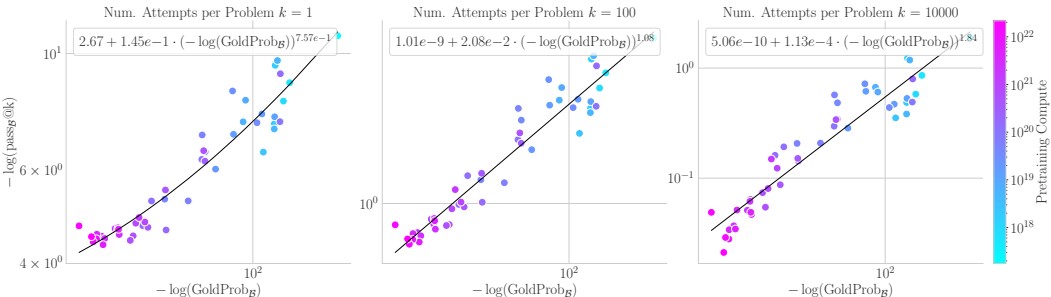

Figure 6: **Scaling of GSM8K Pass Rates with Gold Reference Likelihood (Full Fit).** Each panel fits Eqn. 6, $-\log(\text{pass}_{\mathcal{B}}@k) = \xi_0 + K_0(k) \cdot [-\log(\text{GoldProb}_{\mathcal{B}})]^{\kappa(k)}$. Regressing the gold reference log-likelihoods against the pass rates produces a visually tighter fit with less residual scatter than the compute scaling law (cf. Fig. 1). As $k$ increases, the irreducible error $\xi_0(k)$ falls toward zero and the exponent $\kappa(k)$ increases, making the relationship a steeper, purer power law.

checkpoints exhibit comparatively large relative error, suggesting that while this law better explains the performance of cheaper models, it may struggle with extrapolation to the frontier.

## 4.2 Predicting Pass Rates from Model Parameters and Pretraining Tokens

We evaluated how predictive the parameters + tokens scaling law is using the same backtesting approach (Section 3.2). We again used Pythia 12B model trained for 300B tokens as the target, and iteratively fit Eqn. 4. The relative error in predicting the target model's performance decreases and then plateaus as more capable models are included in the fit (Fig. 5, Top Left). For all values of $k$, these extrapolations become reliable once the largest checkpoint used for fitting is within approximately $\sim 2 - 2.5$ orders of magnitude of the target's compute. Similarly, the five backtested scaling parameters converge to their full-fit values once models within this same compute range of $\sim 1.5 - 2.5$ orders of magnitude are included (Fig. 5, Other Five Panels). While decomposing pretraining compute into parameters and tokens provides a tighter in-range fit, doing so does not yield a significant improvement in predicting the performance of the most expensive model. For small $k$, the parameters+tokens scaling law has slightly lower relative error at predicting the most expensive model's performance, but the advantage disappears for larger $k$ (Fig. 5 Upper Left).

## 5 Fitting and Predicting Pass Rates from Gold Reference Likelihoods

Generative evaluations oftentimes contain not just problems and correct answers, but also "gold references", high quality responses that reach the correct answer for the given problem. We calculated the average log-likelihood of these gold reference sequences to use to predict pass rates:

$$\text{GoldProb}_{\mathcal{B}} \quad \stackrel{\text{def}}{=} \quad \frac{1}{|\mathcal{B}|} \sum_{i \in \mathcal{B}} p_\theta(\text{Gold Reference}_i | \text{Problem}_i). \tag{5}$$

For our gold reference likelihood scaling laws we assessed how well the benchmark $\text{pass}_{\mathcal{B}}@k$ can be (1) fit and (2) predicted as a function of the log likelihoods of the gold references.

### 5.1 Fitting Pass Rates from Gold Reference Likelihoods

Motivated by Schaeffer et al. (2024b), we tested whether this average likelihood of the gold responses can be accurately regressed to fit and predict benchmark pass rates $\text{pass}_{\mathcal{B}}@k$.

$$-\log(\text{pass}_{\mathcal{B}}@k)(\text{GoldProb}_{\mathcal{B}}, k) = \xi_0(k) + K_0(k) \cdot \left[-\log(\text{GoldProb}_{\mathcal{B}})\right]^{\kappa(k)} \tag{6}$$

We find that this straightforwardly computable quantity serves as an exceptionally accurate predictor of benchmark pass rates. As shown in Figure 6, regressing against the negative log likelihood of

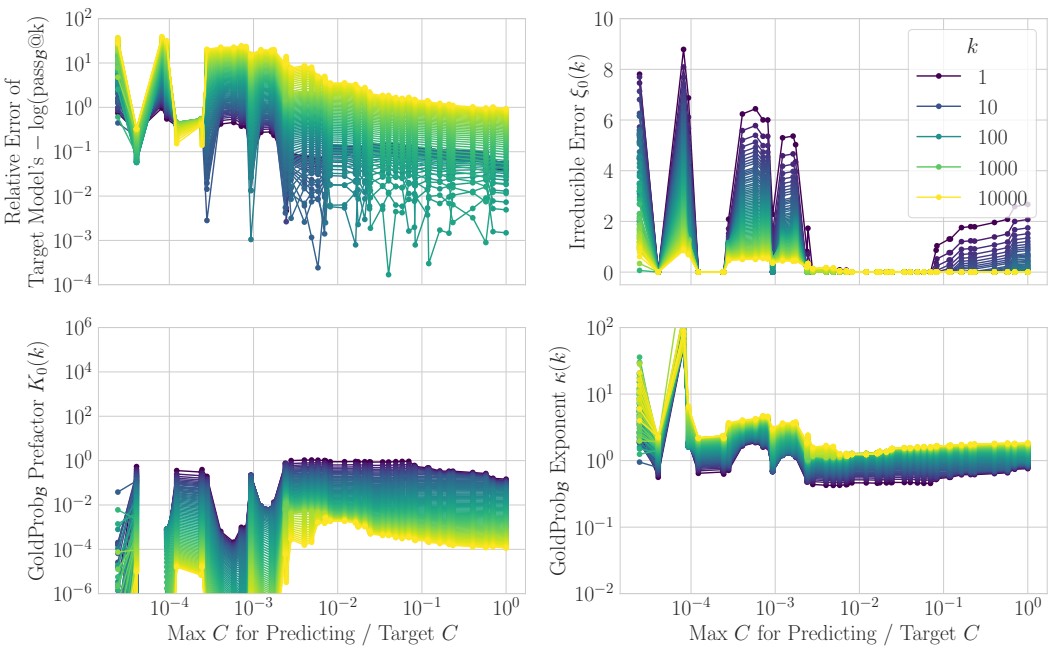

Figure 7: **Predicting GSM8K Pass Rates from Gold Reference Likelihoods (Backtesting). Top Left:** While the overall predictive error is comparable to the other laws, it is slightly lower for small $k$ and slightly higher for large $k$. **Other Three Panels:** The most striking result is the stability of the scaling law's parameters. In stark contrast to the compute and parameters+tokens laws, whose parameters stabilized only when fit on models within $\sim 2$ orders of magnitude of the target, the backtested estimates for $\widehat{\xi}_0(k)$, $\widehat{K}_0(k)$, and $\widehat{\kappa}(k)$ converge to their final values using models up to $\sim 5$ orders of magnitude cheaper than the target. This remarkable stability suggests that the relationship between gold reference likelihoods and pass rates provides a robust signal for long-range forecasts.

the gold references produces a visually tighter fit than the pretraining compute law (Fig. 1). The residual scatter is reduced, and the relationship between $-\log(\text{pass}_{\mathcal{B}}@k)$ and $-\log(\text{GoldProb}_{\mathcal{B}})$ has a smaller reducible error and more closely resembles a power law. We next examined how the number of attempts per problem $k$ influences the scaling law's parameters (Fig. 13). Consistent with our findings for the prior two scaling laws, the irreducible error term $\xi_0(k)$ collapses toward zero as $k$ increases and the exponent $\kappa(k)$ increases smoothly.

## 5.2 PREDICTING PASS RATES FROM GOLD REFERENCE LIKELIHOODS

Using our backtesting approach once more, we evaluated how well the gold references scaling law can predict the performance of our most expensive model. For small $k$, the gold reference likelihoods scaling law has slightly lower relative error at predicting the most expensive model's performance, but worsens for larger $k$ (Fig. 7 Upper Left). Remarkably, the gold reference law exhibits exceptional parameter stability. Unlike compute or parameter+token scaling laws, the backtested estimates for the gold reference scaling parameters stabilize when fitting on models up to $\sim 5$ of magnitude cheaper than the target (Fig. 7, Other Three Panels) and for all values of $k$, suggesting this relationship is a robust signal for long-range forecasting.

## 6 HOW THE COMPUTE LAW EMERGES FROM THE PARAMETERS + TOKENS LAW, WITH IMPLICATIONS FOR OVERTRAINING

The approximately equivalent performance between the compute scaling law (Eqn. 3) and the parameters + tokens scaling law (Eqn. 4) led us to ask whether a deeper connection might exist. We

discovered that *the compute scaling law is the compute-optimal envelope of the parameters+tokens scaling law*. We summarize the insight here; see Appendix E for a deeper discussion.

Fix benchmark $\mathcal{B}$ and attempts per problem $k$. Consider a compute budget $C \approx c\,ND$. The best achievable error is obtained by minimizing the right-hand side of Eqn. 4 over all $(N, D)$ with $ND = C/c$. Evaluating at the minimizer yields Eqn. 3. The mapping of exponents and constants is:

$$\alpha(k) = \left( \tfrac{1}{\beta(k)} + \tfrac{1}{\gamma(k)} \right)^{-1}, \qquad E_0(k) = \mathcal{E}_0(k).$$

**Compute-Optimal Allocation:** Along the envelope (i.e., when Eqn. 3 is tight), the optimal split of compute between parameters and tokens is

$$N^*(C, k) = \left( \tfrac{\beta\,N_0(k)}{\gamma\,D_0(k)} \right)^{\frac{1}{\beta+\gamma}} \left( \tfrac{C}{c} \right)^{\frac{\gamma}{\beta+\gamma}}, \qquad D^*(C, k) = \left( \tfrac{\gamma\,D_0(k)}{\beta\,N_0(k)} \right)^{\frac{1}{\beta+\gamma}} \left( \tfrac{C}{c} \right)^{\frac{\beta}{\beta+\gamma}}.$$

Thus the compute-optimal tokens and parameter *shift with scale* according to $\beta(k), \gamma(k)$; when $\beta(k) = \gamma(k)$, both grow like $C^{1/2}$.

**Departures from the Envelope:** Deviating from the compute-optimal allocation $(N^*, D^*)$ introduces a *multiplicative* penalty $\Phi(r) \geq 1$, where $r(C) = \frac{D^*(C)}{D} = \frac{N}{N^*(C)}$ is the misallocation ratio:

$$-\log\big(\text{pass}_\mathcal{B}@k\big)(C, k) = E_0(k) + C_0(k) \cdot C^{-\alpha(k)} \cdot \underbrace{\left[ \tfrac{\gamma}{\beta+\gamma}\, r^{-\beta} + \tfrac{\beta}{\beta+\gamma}\, r^\gamma \right]}_{\Phi(r;\,\beta(k),\gamma(k))\ \geq 1},$$

with equality iff $r \equiv 1$. Intuitively, $\Phi(r)$ represents a "tax" on performance caused by inefficient compute allocation: small deviations are benign ($\Phi = 1 + \frac{\beta\gamma}{2}(\log r)^2 + O(\log r)^3$ ), but persistent off-ridge scaling significantly reduces "effective" pretraining compute:

$$\frac{\text{Effective Pretraining Compute}}{\text{Pretraining Compute}} = \frac{C(r = 1)}{C(r \neq 1)} \quad (7)$$
$$= \Phi(r;\beta(k),\gamma(k))^{-1/\alpha(k)}$$

A useful case: if one scales $N$ and $D$ at a *fixed* ratio (i.e., without re-optimizing with $C$), the effective compute slope degrades to $\alpha_{\text{path}} = \frac{\min\{\beta,\gamma\}}{2}$.

**Takeaways:** (i) The compute law is not an independent phenomenology; it is the optimal-allocation shadow of the parameters + tokens law. (ii) Use the above allocation to stay on-envelope to maximize compute efficiency. (iii) Implementation constants relating $C$ to $ND$ shift $C_0(k)$, but do not change $\alpha(k)$. All statements hold pointwise in $k$.

**Relationship to Overtraining:** Our misallocation ratio $r(C)$ is the inverse of the overtraining multiplier $M$ studied by Gadre et al. (2024). Our results extend their overtraining scaling law to generative evaluations and the downstream metric $\text{pass}_\mathcal{B}@k$ for $k \geq 1$.

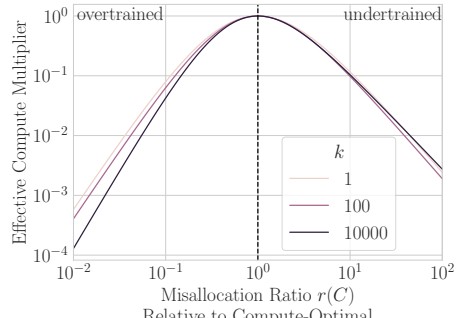

Figure 8: **Quantifying the Cost of Non-Optimal Scaling.** We plot the "effective" compute multiplier (derived in Eq. 7) against the misallocation ratio $r(C)$. The peak at $r = 1$ corresponds to the compute-optimal allocation; deviating from this optimum—either by overtraining ($r < 1$, left) or undertraining ($r > 1$, right) rapidly reduces the effective compute by orders of magnitude. Benchmark: GSM8K.

## 7 HOW SCALING LAWS DEPEND ON THE BENCHMARK: MATH VS GSM8K

We repeated the same scaling analyses for MATH (Hendrycks et al., 2021) as for GSM8K. Due to space constraints, we defer the detailed results to Appendix F and focus here on the salient similarities and differences between the benchmarks.

First, **irreducible error reflects benchmark hardness.** For GSM8K, the irreducible error $E_0(k)$ vanishes quickly, approaching zero by $k \approx 100$. In contrast, MATH retains significant irreducible

error even at high sample counts: $E_0(k) \approx 0.45$ at $k = 10^4$ (Figure 14). This quantifies the intuition that MATH is a significantly harder benchmark.

Second, **harder benchmarks yield steeper power laws at high sample counts.** While both benchmarks show that increasing $k$ steepens the scaling curve, MATH exhibits a more dramatic shift. The compute exponent $\alpha(k)$ for GSM8K rises to $\approx 0.38$ at $k = 10^4$, whereas for MATH it rises to $\approx 0.58$. Because GSM8K performance saturates near 100% coverage, the returns on compute diminish. Conversely, because MATH performance remains far from perfect, the "headroom" allows for steeper scaling returns as compute increases.

Third, **scaling exponents for MATH are sometimes more stable and sometimes less stable when backtesting.** The compute-only scaling law's $\alpha(k)$ and the parameters+tokens's $\beta(k)$ stabilize across 3 orders of magnitude of compute, but the gold reference likelihood's $\kappa(k)$ stabilizes across 2 order of magnitude and the parameters+tokens's $\gamma(k)$ stabilizes across 1 order of magnitude. We do not know what exactly determines this - the hardness of the task, the distribution shift between training data and the task, or some other factor.

.

# 8 DISCUSSION

**Key Findings:** We investigated pretraining scaling laws for generative evaluations of language models. Our work was motivated by OpenAI et al. (2024a)'s success, as well as by the insights of Hu et al. (2024). Our primary contribution is the proposal and rigorous backtesting of three distinct scaling laws for pass-at-$k$, utilizing (1) pretraining compute, (2) model parameters and pretraining tokens, and (3) gold reference log likelihoods as covariates. We demonstrate that the number of attempts per problem, $k$, acts as a critical hyperparameter in generative evaluation, offering a novel lever to control both the parameters of the scaling laws and their predictive reliability. Empirically, while all three laws exhibit comparable predictive accuracy, the gold reference likelihood law is uniquely stable, with parameters converging across nearly five orders of magnitude of compute. Finally, we establish a theoretical connection between the compute law and the parameters-and-tokens law, proving that the former emerges as the compute-optimal envelope of the latter. This derivation yields a dimensionless *misallocation penalty*, quantifying the efficiency loss when training deviates from the optimal parameter-token ratio.

**Limitations:** The primary limitation of this work is its empirical focus on a single model family (Pythia). As detailed in Section 2, this scope was a necessary constraint to enable a rigorous, high-resolution analysis (fitting 5-parameter laws requires dense checkpoints) while managing the substantial computational costs of generative sampling. Additionally, we focus exclusively on pass-at-$k$ metrics; other generative evaluation metrics and tasks warrant similar investigation.

**Related Work:** We defer a detailed discussion of related work to Appendix A.

**Future Directions**: Several avenues for future research are clear. First, subsequent work should determine whether predictive performance can be further improved using controlled pretrained models or more extensive sampling budgets. Second, OpenAI et al. (2024a) suggest that subsetting and/or stratifying problems enables better predictions; future work should isolate exactly which techniques yield the highest signal-to-noise. Third, the strong predictive power of gold reference log-likelihoods warrants theoretical investigation: given that next-token sampling is a branching process with likely exponentially many valid solution paths, it is not immediately obvious why the likelihood of the specific benchmark-provided gold reference correlates so strongly with the pass rate. Two key questions arise: (i) how generalizable is this relationship across diverse benchmarks? and (ii) to what extent does this signal remain robust under heavy optimization pressure? Fourth, while we held decoding strategies constant to isolate the effect of $k$, understanding how different sampling algorithms (e.g., top-$p$) and temperature interact with scaling laws remains an open and valuable question.

ACKNOWLEDGMENTS

RS acknowledges support from Stanford Data Science and from the OpenAI Superalignment Fast Grant. NL is supported by the EPFL AI4science/AI Center program. BM acknowledge support by Schmidt Sciences, Stanford EDGE Scholar Fellowship. SK acknowledges support by NSF 2046795 and 2205329, the MacArthur Foundation, Stanford HAI, OpenAI and Google Inc.

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

## A  RELATED WORK

The study of neural scaling laws, which characterize how model performance predictably improves with resources, has become a cornerstone of modern large-scale machine learning (Hestness et al., 2017; Kaplan et al., 2020a; Hoffmann et al., 2022a; Roberts et al., 2022; Bahri et al., 2024). Foundational work demonstrated that pretraining loss follows a power law with respect to model size, dataset size, and training compute (Kaplan et al., 2020a; Henighan et al., 2020; Hoffmann et al., 2022a). These principles now guide the development of nearly all state-of-the-art language models (OpenAI et al., 2024a; Team et al., 2024; Grattafiori et al., 2024; DeepSeek-AI et al., 2025b; Qwen et al., 2025; Zhipu-AI et al., 2025). This research area has since expanded to explore scaling properties in diverse domains, including computer vision (Zhai et al., 2022), machine translation (Ghorbani et al., 2021), and even for optimizing inference-time compute rather than training (Wu et al., 2024; Snell et al., 2024). Despite this breadth, scaling laws for complex, multi-step *generative tasks*—which often represent the ultimate goal of language modeling—have remained comparatively underexplored due to the challenges of evaluation. Our work builds most directly on recent efforts to forecast performance on such generative evaluations. The *GPT-4 Technical Report* pioneered this direction by showing that the negative log pass rate on a subset of the HumanEval coding benchmark scaled predictably as a power law of pretraining compute (OpenAI et al., 2024a); they successfully predicted GPT-4's performance by extrapolating from smaller models over approximately three orders of magnitude (OpenAI et al., 2024a). We adopt their proposed functional form for our compute-based law and extend their work by systematically analyzing how the number of attempts, $k$, reshapes the law's parameters. Furthermore, our rigorous backtesting reveals the brittleness of this compute-only law, finding it reliable only across 1-2 orders of magnitude in our setting. Separately, our work shares a philosophical motivation with Hu et al. (2024), who investigate "infinite resolution" sampling for emergent abilities.

**Relationship Between Pass-at-$k$ and Accuracy:** In the context of language model evaluations, accuracy and pass@$k$ are fundamentally different. "Accuracy" is a likelihood comparison: does the correct multiple-choice answer have the highest probability relative to the incorrect alternative answers? (Gao et al., 2024; Schaeffer et al., 2024a). In contrast, pass@k measures the success of free-form generative sampling. To make the distinction clear, here are three obvious differences:

1. Accuracy is either binary 0 or 1, whereas pass@$k$ is a continuous quantity in $[0, 1]$ and practically almost always in $(0, 1)$. The two metrics have different output spaces.

2. As functions, the two metrics have different inputs: accuracy is a function of the correct answer and the incorrect answer, whereas pass@k does not depend on the incorrect answer.

3. Accuracy is computed using the exact sequence of the correct answer via teacher forcing. In comparison, pass@$k$ can sample any sequence of tokens.

**Detailed Comparison with Gadre et al. (2024):**  On the empirical side, the 46 downstream tasks in Gadre et al. (2024) fall into two categories (see their Table 5): (1) pretraining losses (i.e., next-token prediction), (2) discriminative evaluations: measuring accuracy on multiple-choice tasks (e.g., MMLU). Our work focuses on a third, distinct category: generative evaluation, scored with pass@$k$.

On the theoretical side, our starting point is the standard separable $((N, D))$ scaling law of Hoffmann et al. (2022a) ($L(N, D) \approx E + AN^{-\alpha} + BD^{-\beta}$). Gadre et al. (2024) show in their Appendix B that this form implies a compute power law with exponent $(\alpha\beta/(\alpha + \beta))$, and that departures from the compute-optimal allocation modify only the prefactor as a function of an over-training parameter (m). Our derivation in Appendix E is a direct generalization of this result to the relevant quantity for inference scaling ($-\log \text{pass}@k$) and to arbitrary $(k)$: we retain the same $((N, D))$ ansatz, derive the compute-optimal envelope and its exponent $(\alpha(k) = (1/\beta(k) + 1/\gamma(k))^{-1})$, and explicitly express how sub-optimal choices of $((N, D))$ enter as a multiplicative penalty depending on the ratio $(N/N^*)$.

## B    METHODOLOGY: NUMBER OF SAMPLES PER PROBLEM PER MODEL

Because sampling from language models is expensive and because our compute budget is limited, we chose a compute-aware sampling strategy designed to balance resolution with cost. A key consideration is that the number of samples per problem sets the resolution, i.e., if we draw $n$ samples per problem, then any pass rate on that problem below $1/n$ will likely appear to be $0$.

For each model checkpoint, we took the following strategy: we drew a minimum of $2^{14}$ samples per model per problem, and then continued sampling until 10 successes were obtained or until a maximum of $2^{15}$ samples were drawn. $2^{14}$ was chosen so that we would have more samples per problem than the largest number of attempts per problem $k$ we considered ($1e4$).

Fig. 9 shows how many samples per GSM8K problem per model we had at the time our analyses were conducted, and Fig. 10 shows how many samples per MATH problem per model we had at the time our analyses were conducted. In total, we had $\sim 500M$ total samples for GSM8K and $\sim 400M$ total samples for MATH. Additional sampling would have likely benefited our analyses.

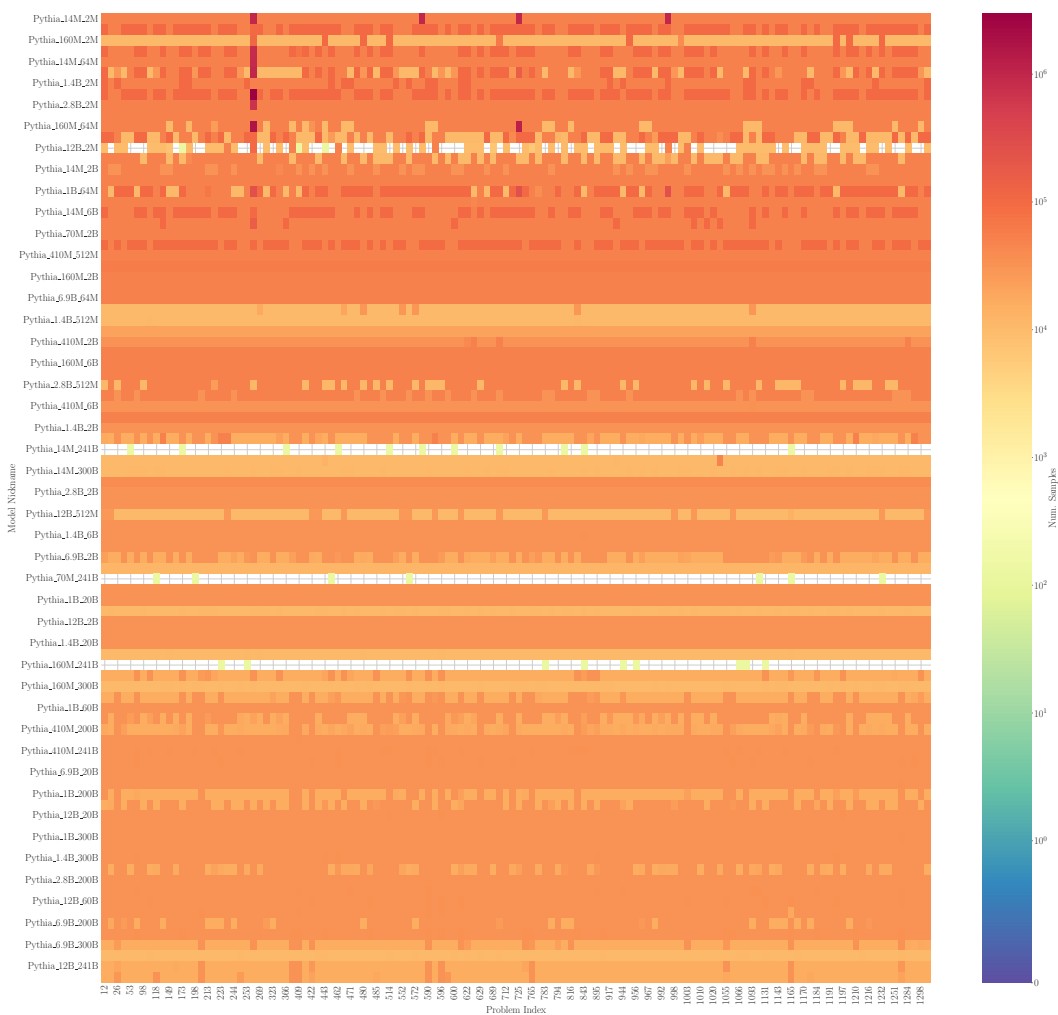

Figure 9: **Number of Samples per Model per GSM8K Problem.** This heatmap visualizes the number of samples per model (y-axis) per problem (x-axis) at the time our analyses were conducted.

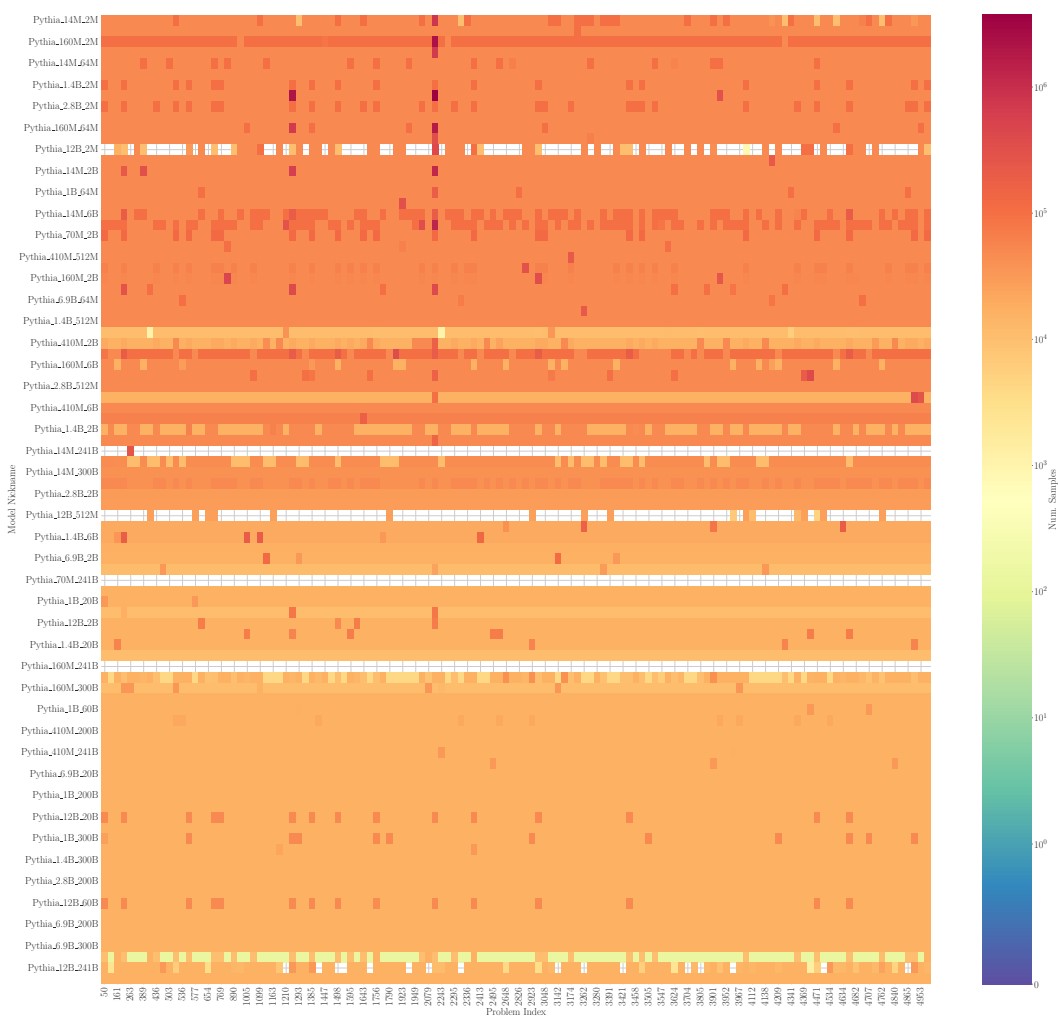

Figure 10: **Number of Samples per Model per MATH Problem.** This heatmap visualizes the number of samples per model (y-axis) per problem (x-axis) at the time our analyses were conducted.

## C   ESTIMATION OF TOTAL SAMPLING COST

Sampling was performed using vLLM (Kwon et al., 2023). We generated 128 samples of the same problem at a time to make use of prefix caching for more efficient sampling.

### C.1   ESTIMATION OF GSM8K SAMPLING COSTS

Estimates of the GPU time per 128 samples per model size for GSM8K are provided in Table 1.

| Pythia Model Parameters | A100-s Per 128 Samples (Approximate) |
|---|---|
| 14M | 4 |
| 70M | 4 |
| 160M | 5 |
| 410M | 7 |
| 1B | 9 |
| 1.4B | 11 |
| 2.8B | 13 |
| 6.9B | 25 |
| 12B | 36 |

Table 1: **A100-Seconds Per 128 Samples by Model Size on GSM8K Problems.**

At a minimum of $3.2e4$ samples per problem, 128 problems per model checkpoint, 8 model checkpoints per model size, our total compute costs are lower bounded by:

$$
\begin{aligned}
\text{Total Chip-Time} &\geq \sum_{\text{Model Size}} \frac{8 \text{ Checkpoints}}{\text{Model Size}} \cdot \frac{128 \text{ Problems}}{1 \text{ Checkpoint}} \cdot \frac{32e4 \text{ Samples}}{1 \text{ Problem}} \cdot \frac{X_{\text{Model Size}} \text{ A100-s}}{128 \text{ Samples}} \\
&\geq 292e6 \text{ A100-s} \\
&\geq 81e3 \text{ A100-hr}
\end{aligned}
$$

Thus, sampling our GSM8K data is lower bounded by $81e3$ A100-hr. To estimate the total financial cost of this sampling, we looked up pricing of compute providers. As of November 2025, Lambda offers A100s at \$1.79/A100-hr, RunPod at \$1.39/A100-hr, Tensordock at \$1.63/A100-hr, and Vast-AI at \$1.40/A100-hr. In total, the financial cost of our GSM8K sampling data is more than $112k$.

### C.2   ESTIMATION OF MATH SAMPLING COST

Estimates of the GPU time per 128 samples per model size for MATH are provided in Table 2.

| Pythia Model Parameters | A100-s Per 128 Samples (Approximate) |
|---|---|
| 14M | 4 |
| 70M | 4 |
| 160M | 5 |
| 410M | 8 |
| 1B | 10 |
| 1.4B | 10 |
| 2.8B | 15 |
| 6.9B | 27 |
| 12B | 39 |

Table 2: **A100-Seconds Per 128 Samples by Model Size on MATH Problems.**

# D   GSM8K Scaling Law Parameters by Number of Attempts Per Problem $k$

For the three scaling laws we consider, we visualize how the fit GSM8K scaling law parameters change as a function of the number of attempts per problem $k$ (Fig. 11, Fig. 12, Fig. 13).

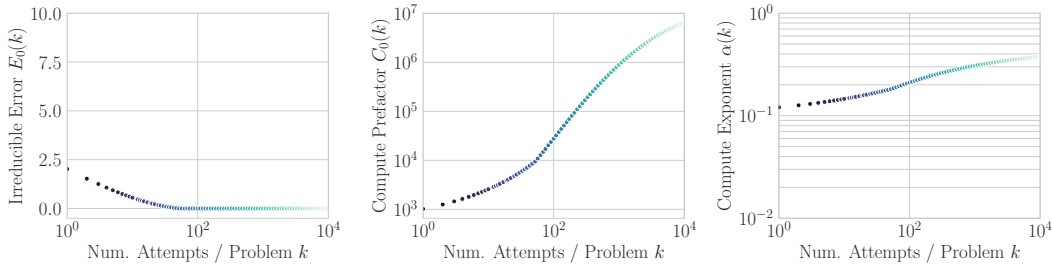

Figure 11: **GSM8K Pretraining Compute Scaling Law Parameters by Number of Attempts per Problem (Full Fits).** Scaling law parameters of Eqn. 3 as functions of $k$. Hue corresponds to $k$.

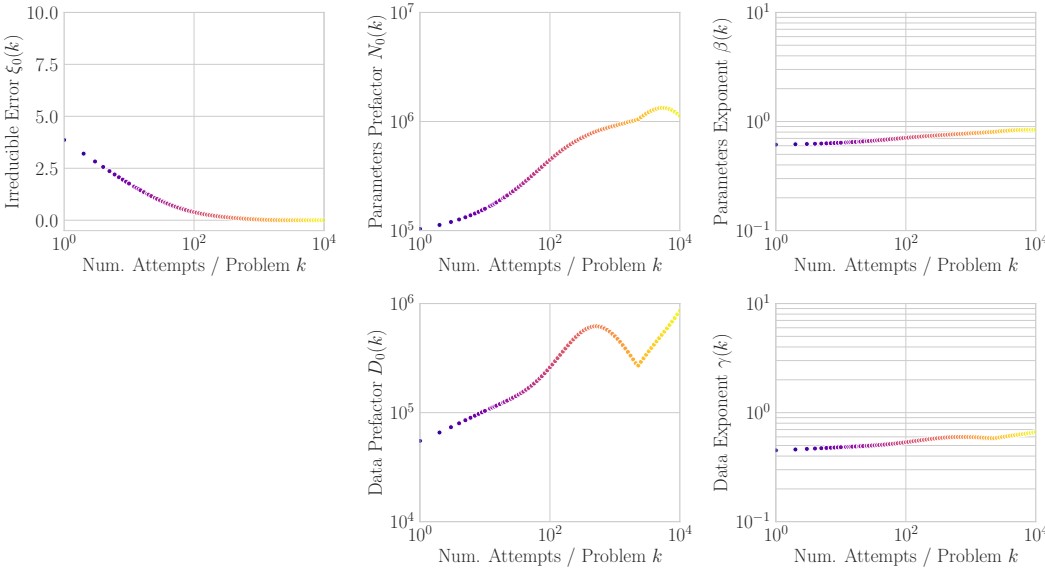

Figure 12: **GSM8K Parameters and Tokens Scaling Law Parameters by Number of Attempts per Problem (Full Fits).** Scaling law parameters of Eqn. 4 as functions of $k$. Hue corresponds to $k$.

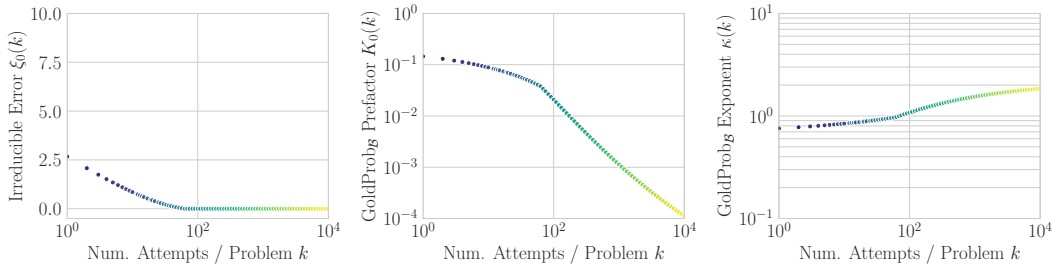

Figure 13: **GSM8K Gold Reference Log Likelihoods Scaling Law Parameters by Number of Attempts per Problem (Full Fits).** Scaling law parameters of Eqn. 5 as functions of $k$. Hue corresponds to $k$.

# E RELATION BETWEEN THE COMPUTE SCALING LAW AND THE PARAMETERS + TOKENS SCALING LAW

Here, we provide a concrete analytic relation between Eqn. 3 and Eqn. 4 provided in the main text. For a fixed benchmark $\mathcal{B}$ and number of attempts per problem $k$, we model

$$-\log\big(\text{pass}_\mathcal{B}\,@k\big)(N, D, k) = \mathcal{E}_0(k) + \frac{N_0(k)}{N^{\beta(k)}} + \frac{D_0(k)}{D^{\gamma(k)}}, \tag{8}$$

and, when training under a compute budget $C = c\,ND$,

$$-\log\big(\text{pass}_\mathcal{B}\,@k\big)(C, k) = E(k) + \frac{C_0(k)}{C^{\alpha(k)}}. \tag{9}$$

**Compute-optimal envelope.** Assume the excess loss *beyond the irreducible term* is additive and separable in $(N, D)$ as in Eq. 8, with $N_0(k), D_0(k) > 0$ and exponents $\beta(k), \gamma(k) > 0$. For each fixed $k$, define

$$A = N_0(k), \quad B = D_0(k), \quad \beta = \beta(k), \quad \gamma = \gamma(k), \tag{10}$$

and

$$F(N, D) = \frac{A}{N^\beta} + \frac{B}{D^\gamma}, \qquad \mathcal{F}(C) = \min_{\substack{N, D > 0: \\ cND = C}} F(N, D). \tag{11}$$

Then the compute law Eq. 9 is the compute-optimal envelope of Eq. 8, with

$$E(k) = \mathcal{E}_0(k), \qquad \alpha(k) = \frac{\beta(k)\,\gamma(k)}{\beta(k) + \gamma(k)} = \left(\frac{1}{\beta(k)} + \frac{1}{\gamma(k)}\right)^{-1}, \tag{12}$$

$$C_0(k) = (\beta + \gamma)\,\beta^{-\frac{\beta}{\beta+\gamma}}\,\gamma^{-\frac{\gamma}{\beta+\gamma}}\,A^{\frac{\gamma}{\beta+\gamma}}\,B^{\frac{\beta}{\beta+\gamma}}\,c^{\alpha(k)}. \tag{13}$$

The compute-optimal allocation along the envelope is

$$N^*(C, k) = \left(\frac{\beta A}{\gamma B}\right)^{\frac{1}{\beta+\gamma}}\left(\frac{C}{c}\right)^{\frac{\gamma}{\beta+\gamma}}, \qquad D^*(C, k) = \left(\frac{\gamma B}{\beta A}\right)^{\frac{1}{\beta+\gamma}}\left(\frac{C}{c}\right)^{\frac{\beta}{\beta+\gamma}}. \tag{14}$$

**Derivation Sketch.** Form the Lagrangian $\mathcal{L} = AN^{-\beta} + BD^{-\gamma} + \lambda\,(ND - C/c)$. First-order conditions give $\beta AN^{-\beta} = \gamma BD^{-\gamma}$ and $ND = C/c$, from which (A.5) follows. Substituting $N^*, D^*$ into $F(N, D)$ yields $\mathcal{F}(C) = C_0(k)\,C^{-\alpha(k)}$ with $\alpha(k)$ and $C_0(k)$ as in Eqs. 12-13, and $E(k) = \mathcal{E}_0(k)$.

**Remarks.**

1. The compute exponent is the parallel sum of the $(N, D)$ exponents: $\alpha = (1/\beta + 1/\gamma)^{-1}$.

2. If $\beta = \gamma$, then $\alpha = \beta/2$ and $N^*, D^* \propto C^{1/2}$ with $C_0 = 2\,\beta^{-1}(AB)^{1/2}c^{\beta/2}$.

3. If one scales $N$ and $D$ at a fixed ratio (i.e., without re-optimizing with $C$), the effective compute exponent degrades to $\alpha_{\text{path}} = \min\{\beta, \gamma\}/2$.

4. All quantities may depend on $k$; the mapping in Eqs. 12-14 applies point-wise in $k$. The conversion constant $c$ enters only via $C_0$, not $\alpha$.

## E.1 MULTIPLICATIVE DEVIATION FROM THE OPTIMAL COMPUTE ENVELOPE

Let $(N^*(C), D^*(C))$ denote the compute–optimal allocation in Eq. 14 and define the *misallocation ratio*

$$r(C) := \frac{N(C)}{N^*(C)} = \frac{D^*(C)}{D(C)}. \tag{15}$$

A direct algebraic manipulation using the first-order optimality condition $\beta A(N^*)^{-\beta} = \gamma B(D^*)^{-\gamma}$ gives the exact factorization

$$\frac{-\log\big(\text{pass}_{\mathcal{B}} @k\big)(C) - \mathcal{E}_0}{-\log\big(\text{pass}_{\mathcal{B}} @k\big)_{\text{opt}}(C) - \mathcal{E}_0} = \Phi(r; \beta, \gamma) := \frac{\gamma}{\beta + \gamma}\, r^{-\beta} + \frac{\beta}{\beta + \gamma}\, r^{\gamma}\ \geq 1, \tag{16}$$

with equality iff $r \equiv 1$. Equivalently, this produces a multiplicative correction to Eq. 9

$$-\log(\text{pass}_{\mathcal{B}} @k)(C) = \mathcal{E}_0 + C_0\, C^{-\alpha}\ \underbrace{\Phi\big(\tfrac{N}{N^*}; \beta, \gamma\big)}_{\text{misallocation factor } \geq 1}. \tag{17}$$

Thus the deviation from equality is *fully parameterized* by $r$ via the dimensionless penalty $\Phi \geq 1$.

**Local and asymptotic behavior of the penalty.** Let $t = \log r$. A Taylor expansion of equation 16 around $r = 1$ gives

$$\Phi(r; \beta, \gamma) = 1 + \frac{\beta\gamma}{2}\, t^2 + O(t^3), \tag{18}$$

so small misallocations incur only a *quadratic* penalty in log-space. Along a power-law path $N \propto C^p$ (with constants absorbed into $r$), $t = \big(p - \frac{\gamma}{\beta+\gamma}\big) \log C + O(1)$; hence if $p \neq \frac{\gamma}{\beta+\gamma}$ the penalty grows polynomially:

$$\Phi(r(C)) \asymp \begin{cases} \frac{\beta}{\beta+\gamma}\, r(C)^{\gamma}, & r(C) \to \infty \quad \text{(over-allocating } N\text{)}, \\ \frac{\gamma}{\beta+\gamma}\, r(C)^{-\beta}, & r(C) \to 0 \quad \text{(under-allocating } N\text{)}. \end{cases} \tag{19}$$

This isolates the compute–optimal scaling $C^{-\alpha}$ and expresses all off-ridge effects through a *single*, dimensionless factor depending only on the allocation ratio $N/N^*$ (or equivalently $D^*/D$).

We conclude that if one matches the compute law without optimizing $(N, D)$, the result is *not* a single power law in general: it is a compute–optimal power multiplied by a misallocation penalty $\Phi(N/N^*) \geq 1$.

# F  HOW SCALING DEPENDS ON THE BENCHMARK: THREE PRETRAINING SCALING LAWS FOR MATH

In the main text, we presented results for GSM8K (Cobbe et al., 2021). Here, we present corresponding results for MATH (Hendrycks et al., 2021). For a focused discussion of the similarities and differences between the two benchmarks, please see Section 7 in the main text.

## F.1  FITTING AND PREDICTING PASS RATES FROM PRETRAINING COMPUTE

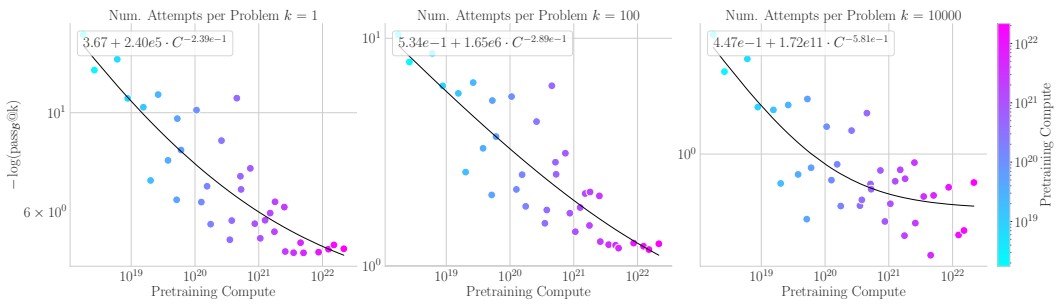

Figure 14: **Scaling of MATH Pass Rates with Pretraining Compute (Full Fit).** Each panel fits Eqn. 3 $-\log\big(\text{pass}_{\mathcal{B}}@k\big)(C,k) = E_0(k)+C_0(k)\cdot C^{-\alpha(k)}$ to MATH pass rates for Pythia checkpoints across $\sim$5 orders of magnitude of pretraining compute for $k \in \{1, 10^2, 10^4\}$.

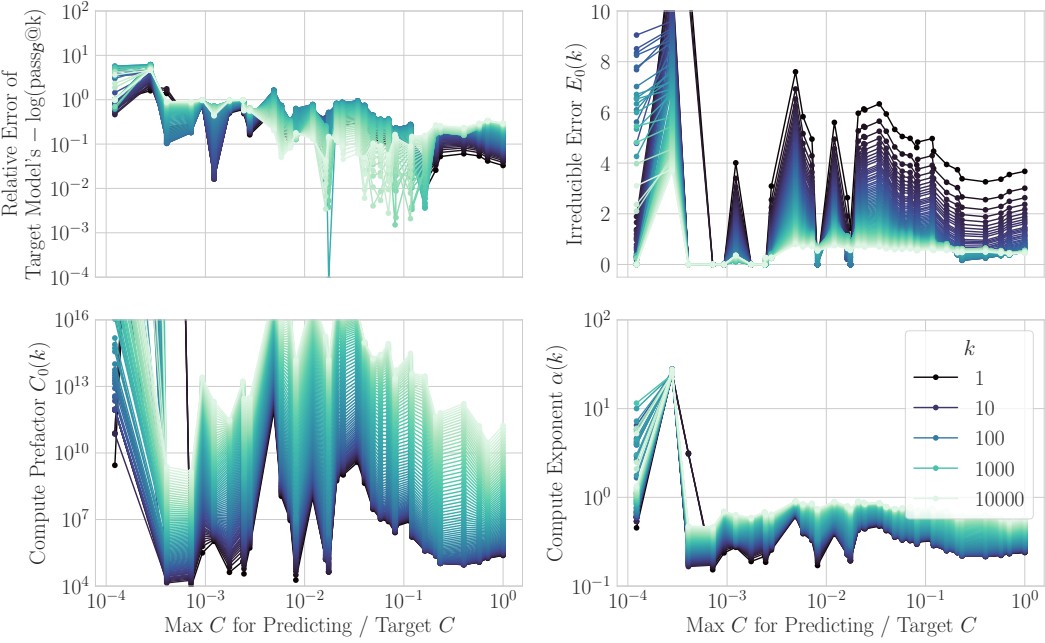

Figure 15: **Predicting MATH Pass Rates from Scaling Pretraining Compute (Backtesting).** We evaluate predictability via *backtesting*: we iteratively fit Eq. 3 on subsets of models up to a maximum pretraining compute, then extrapolate to predict the most expensive model's $-\log\big(\text{pass}_{\mathcal{B}}@k\big)$. The most expensive model is Pythia 12B-parameter 300B-token ($\approx 2.16 \times 10^{22}$ FLOP).

### F.2 FITTING AND PREDICTING PASS RATES FROM PARAMETERS AND TOKENS

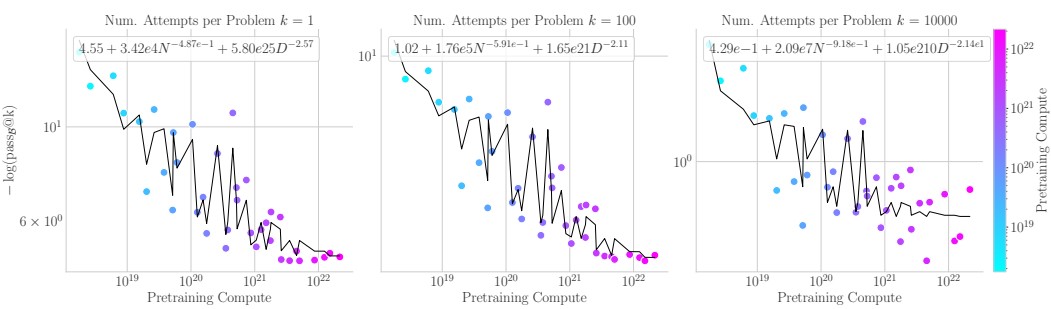

Figure 16: **Scaling of MATH Pass Rates with Parameters and Tokens (Full Fit).** Each panel fits Eqn. 4, $-\log\big(\mathrm{pass}_{\mathcal{B}}@k\big)(N, D, k) = \mathcal{E}_0(k) + N_0(k)\,N^{-\beta(k)} + D_0(k)\,D^{-\gamma(k)}$.

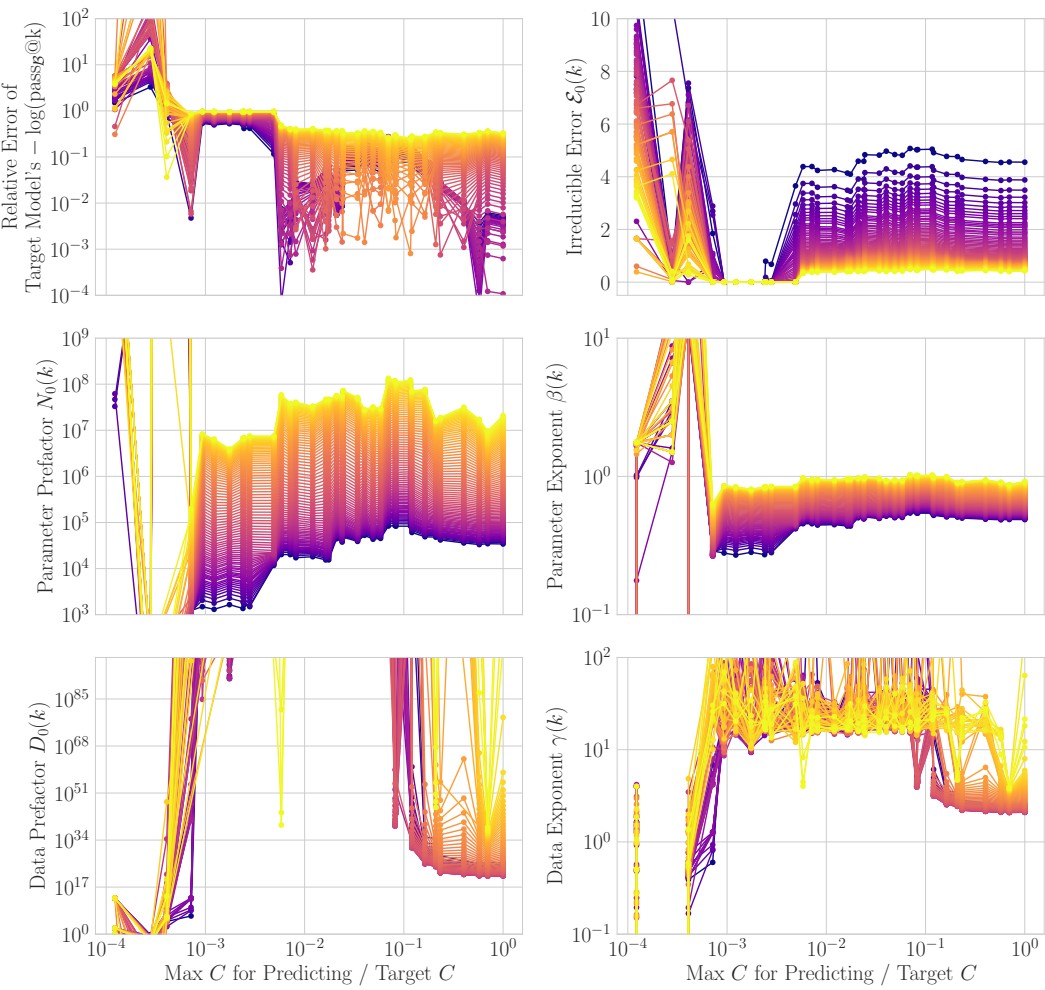

Figure 17: **Predicting MATH Pass Rates from Scaling Parameters and Tokens (Backtesting).** We evaluated how accurately the parameters + tokens scaling law (Eqn. 4) predicts the most expensive model's $-\log(\mathrm{pass}_{\mathcal{B}}@k)$.

### F.3 FITTING AND PREDICTING PASS RATES FROM GOLD REFERENCE LIKELIHOODS

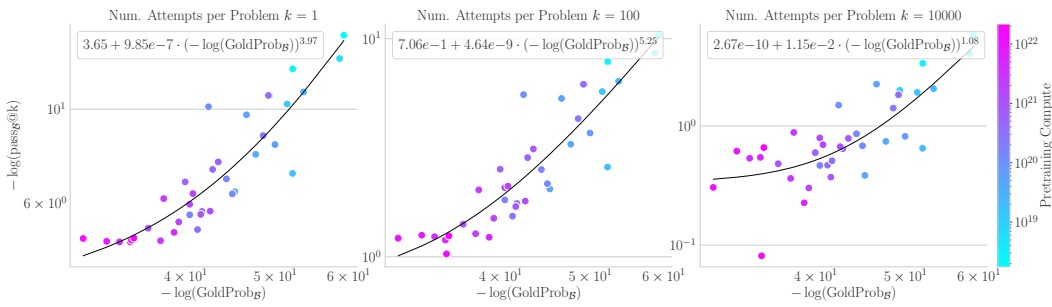

Figure 18: **Scaling of MATH Pass Rates with Gold Reference Likelihood (Full Fit).** Each panel fits Eqn. 6, $-\log(\mathrm{pass}_{\mathcal{B}}@k) = \xi_0 + K_0(k) \cdot \left[-\log\left(\mathrm{GoldProb}_{\mathcal{B}}\right)\right]^{\kappa(k)}$, for $k \in \{1, 10^2, 10^4\}$.

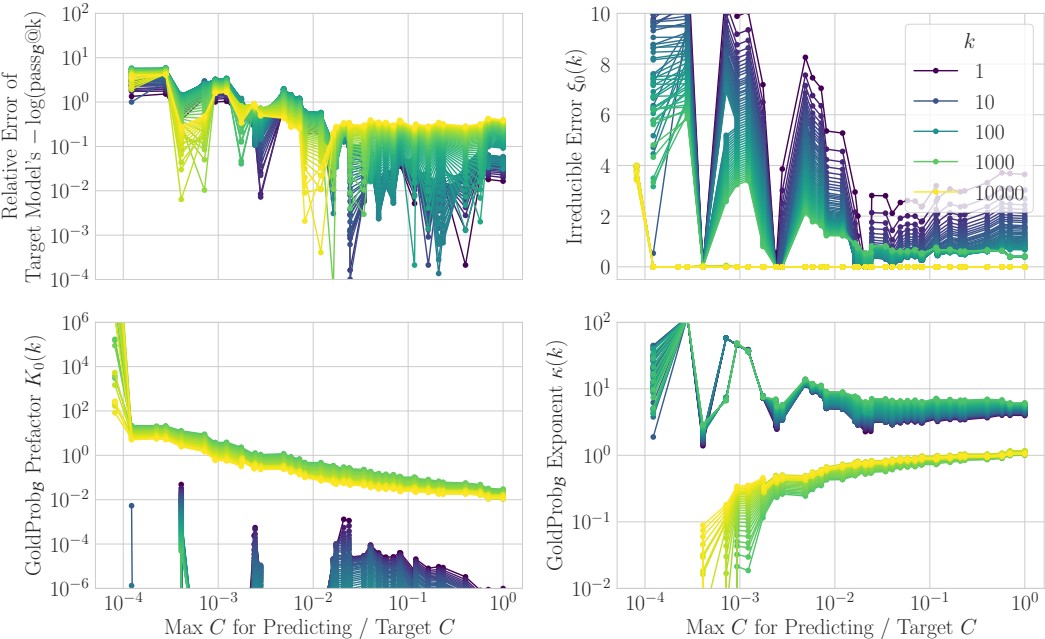

Figure 19: **Predicting MATH Pass Rates from Gold Reference Likelihoods (Backtesting).** We evaluated how accurately the gold reference likelihoods scaling law (Eqn. 6) predicts the most expensive model's $-\log(\mathrm{pass}_{\mathcal{B}}@k)$.

# G  MATH SCALING LAW PARAMETERS BY NUMBER OF ATTEMPTS PER PROBLEM $k$

For the three scaling laws we consider, we visualize how the fit MATH scaling law parameters change as a function of the number of attempts per problem $k$ (Fig. 20, Fig. 21, Fig. 22).

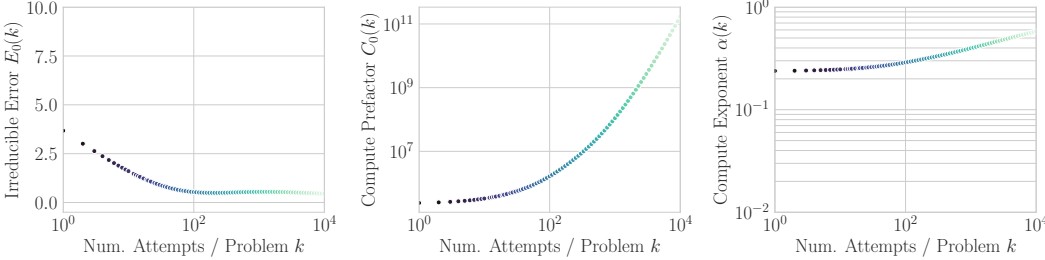

Figure 20: **MATH Pretraining Compute Scaling Law Parameters by Number of Attempts per Problem (Full Fits).** Scaling law parameters of Eqn. 3 as functions of $k$. Hue corresponds to $k$.

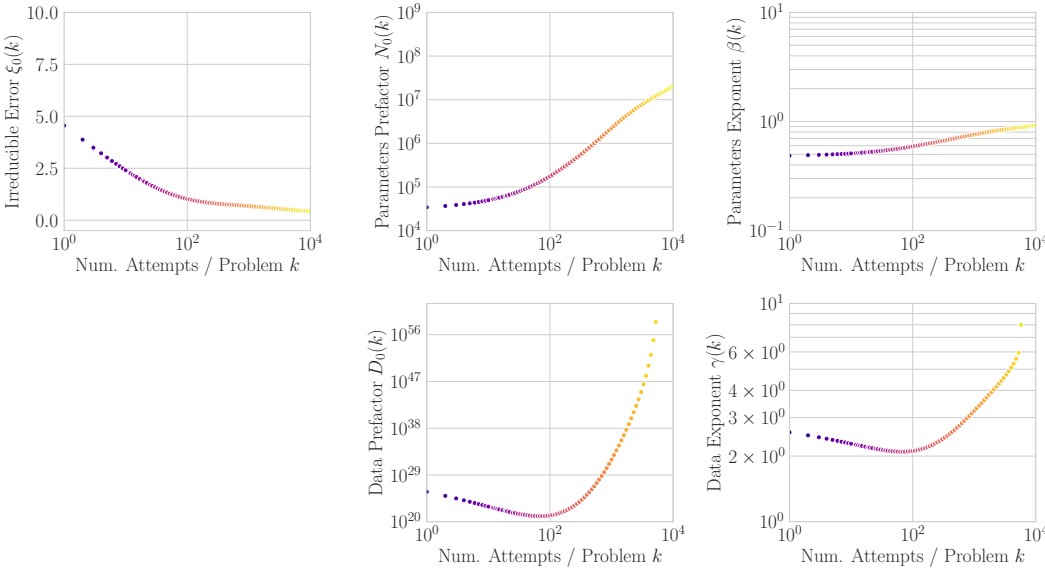

Figure 21: **MATH Parameters and Tokens Scaling Law Parameters by Number of Attempts per Problem (Full Fits).** Scaling law parameters of Eqn. 4 as functions of $k$. Hue corresponds to $k$.

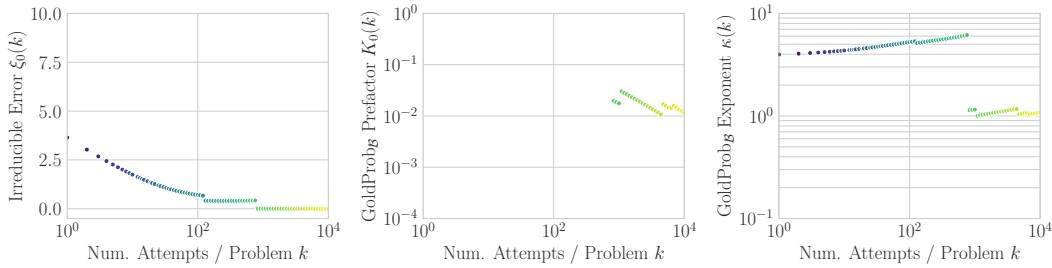

Figure 22: **MATH Gold Reference Log Likelihoods Scaling Law Parameters by Number of Attempts per Problem (Full Fits).** Scaling law parameters of Eqn. 5 as functions of $k$. Hue corresponds to $k$.

