# OpenReview forum: "Pretraining Scaling Laws for Generative Evaluations of Language Models"
_ICLR.cc/2026/Conference — ICLR 2026 Poster_

### Official Review · Reviewer_XVLF · 2025-10-20

**Soundness:** 2
**Presentation:** 2
**Contribution:** 2
**Rating:** 4
**Confidence:** 3

**Summary:**

This paper studies scaling laws for generative evaluations, which is a relatively unexplored area compared to pretraining loss or discriminative benchmarks. The authors propose three types of scaling laws that relate model performance (measured by pass@k) to:
1. Pretraining compute
2. Model parameters and pretraining tokens
3. Log-likelihoods of gold reference solutions

They empirically evaluate these laws on GSM8K using Pythia models (14M-12B parameters, 300B tokens), showing that: All three scaling laws achieve comparable predictive performance.

**Strengths:**

1. The paper expands scaling law studies beyond pretraining or discriminative settings to generative evaluations, which are highly relevant for coding, reasoning, and mathematical tasks.
2. The authors clearly define three scaling formulations, use consistent benchmarks (GSM8K), and employ backtesting to rigorously evaluate predictive ability.
3. The results show stable behavior over large dynamic ranges, and the study identifies when extrapolation becomes unreliable

**Weaknesses:**

1. The writing quality of the paper could be improved. For example, Figures 3 and 5 omit certain points, and the legend partially obscures the plotted lines.
2. All experiments are conducted on GSM8K using Pythia checkpoints
3. The paper focuses only on pass@k with temperature sampling = 1.0. Other decoding strategies (top-p, nucleus sampling, self-consistency) might influence scaling behavior.
4. The results are derived solely from internal checkpoints of a single architecture. Evaluating additional model families such as LLaMA or Qwen could help assess the robustness of the proposed scaling laws. Moreover, since Pythia is a relatively dated model, it may not be well suited for evaluating mathematical reasoning capabilities.
5. I believe that pass@k can be viewed as a variant of accuracy. Could the authors clarify in detail how this work differs from Language Models Scale Reliably with Over-Training and on Downstream Tasks https://arxiv.org/abs/2403.08540.

**Questions:**

See weaknesses.

---

> ### Author Response · Authors · 2025-11-20
> **Response to Reviewer XVLF (Part 1)**
>
> We thank Reviewer XVLF for their thoughtful and constructive feedback. We are encouraged that they found our work on a relevant, underexplored topic to be clearly defined and rigorously evaluated. We agree with the weaknesses they identified, which primarily concern presentation and the scope of our empirical results. In our revision, we will address these points, which we believe significantly strengthen the paper:
>
> # Weakness #1: Improve figures
>
> > Figures 3 and 5 omit certain points, and the legend partially obscures the plotted lines.
>
> We appreciate the reviewer pointing out the presentation issues with Figures 3 and 5. This was an oversight. We will update these figures to ensure all data points are clearly visible and that the legends do not obscure any plotted lines.
>
> > The writing quality of the paper could be improved.
>
> In addition to correcting the figures, we will perform a thorough proofread of the entire manuscript to improve overall clarity.
>
> # Weakness #5: Novelty vs. Gadre et al. (2024)
>
> > I believe that pass@k can be viewed as a variant of accuracy
>
> In the context of language model evaluations, accuracy and pass@k are fundamentally different. The “accuracy” in Gadre et al. is a likelihood comparison (e.g., does the correct multiple-choice answer have the highest probability relative to the incorrect alternative answers?). In contrast, pass@k measures the success of free-form generative sampling. To make the distinction clear, here are three obvious differences:
>
> 1. Accuracy is either binary 0 or 1, whereas pass@k is a continuous quantity in $[0, 1]$ and practically almost always in $(0, 1)$. The two metrics have different output spaces.
>
> 2. As functions, the two metrics have different inputs: accuracy is a function of the correct answer and the incorrect answer, whereas pass@k does not depend on the incorrect answer.
>
> 3. Accuracy is computed using the exact sequence of the correct answer via teacher forcing. In comparison, pass@k can sample any sequence of tokens.
>
> > Could the authors clarify in detail how this work differs from Language Models Scale Reliably with Over-Training and on Downstream Tasks https://arxiv.org/abs/2403.08540.
>
> The 46 downstream tasks in Gadre et al. (2024) fall into two categories (see their Table 5):
>
> 1. Language Modeling: Measuring pretraining loss (i.e., next-token prediction).
> 2. Discriminative Evaluation: Measuring accuracy on multiple-choice tasks (e.g., MMLU).
>
> **Our work introduces a third, distinct category: generative evaluation via sampling**, measured by pass@k. We will better clarify how Gadre motivates our work, e.g., that out of 46 downstream evaluations, not one was generative, motivating our work to study the scaling behavior of generative evaluations.
>
> # Weakness #4: Single model family
>
> > All experiments are conducted [...] using Pythia checkpoints
>
> > The results are derived solely from internal checkpoints of a single architecture. Evaluating additional model families such as LLaMA or Qwen could help assess the robustness of the proposed scaling laws.
>
> We fully agree that evaluating our proposed scaling laws across multiple model families would be the gold standard for assessing their robustness. However, our specific methodology, which involves fitting and rigorously backtesting multi-variable scaling laws, imposes prerequisites that these other families unfortunately do not meet.
>
> Our analysis requires model families to be:
>
> 1. Publicly available.
> 2. Densely sampled across model parameters (N), pretraining tokens (D) and pretraining compute (C).
> 3. Spanning multiple orders of magnitude of pretraining compute.
>
> Most modern, state-of-the-art model families fail to meet these criteria for a scientific study of scaling:
>
> Re: LLaMA 3 & 4: These two families offer three models each, whereas our parameters + tokens scaling law contains five free parameters; robustly fitting 5-parameter scaling law with only 3 data points is dubious.
>
> Re: Qwen: The Qwen 2.5 technical report states a total corpus size (18T), as does Qwen 3 (36T), but neither clearly reports the exact number of tokens seen by each individual model checkpoint. Without this data, we cannot use pretraining tokens (D) or pretraining compute (C) as covariates without making unsupported assumptions about the per-model token budgets.
>
> Thus, the choice of Pythia was a deliberate methodological necessity. This unique dataset allowed us to rigorously fit our models and, most importantly, perform the systematic backtesting (Figs. 3, 5, 7) that is a core contribution of our paper. We will add this clarification to our manuscript.
>
> > Pythia is a relatively dated model, it may not be well suited for evaluating mathematical reasoning capabilities
>
> This is precisely why we chose a slightly older and easier mathematical reasoning benchmark: we wanted to be in a Goldilocks zone where the problems are not too easy and not too hard for the models such that we could identify scaling trends.

---

> ### Author Response · Authors · 2025-11-20
> **Response to Reviewer XVLF (Part 2)**
>
> # Weakness #3: Additional decoding algorithms
>
> > The paper focuses only on pass@k with temperature sampling = 1.0. Other decoding strategies (top-p, nucleus sampling, self-consistency) might influence scaling behavior.
>
> Generative evaluations indeed introduce a vast new space of methodological choices, including the metric, its hyperparameters (like $k$), the sampling algorithm (e.g., nucleus sampling), and its hyperparameters (e.g., $\tau, p$).
>
> This is precisely why we chose a deliberately focused approach. A core contribution of our paper is to demonstrate that these generative-setting choices are powerful levers that can fundamentally alter the scaling law parameters themselves.
>
> Rather than superficially exploring many of these levers, we chose to perform a deep and rigorous analysis on a single, critical one: the number of attempts per problem, k, in the pass@k metric.
>
> Our focus on $k$ was intentional, as it directly relates to the recent and growing interest in scaling inference-time (a.k.a. test-time) compute (e.g., best-of-n sampling, as seen in Brown et al., 2024; Ehrlich et al., 2025). Our work is the first to show how this k-scaling lever interacts with pretraining scaling laws, revealing, for example, that it can eliminate the irreducible error term and steepen the compute exponent.
>
> Regarding other decoding strategies, we had two primary reasons for not including them:
>
> Computational Cost: Generative sampling is expensive. As we quantify in a newly added Appendix C, our experiments already represent a substantial computational cost (over $100k USD). Extending this 5-order-of-magnitude analysis across another axis of hyperparameters was, unfortunately, infeasible.
>
> Impact (Based on Prior Work): Prior work suggests that other parameters, like temperature, may have a simpler, less interesting effect. For instance, Chen et al. (2021)’s Fig. 4 suggests that changing temperature applies a minor affine transformation pass@k.
>
> We have added text to our methodology making this deliberate focus on k explicit and have moved the discussion of other sampling strategies to our Future Work section.
>
> # Weakness #2: Single Benchmark
>
> > All experiments are conducted on GSM8K
>
> We agree with the reviewer that expanding our analysis to more benchmarks is a critical next step for generalizing our findings.
>
> However, generative sampling across this many model checkpoints is exceptionally compute-intensive. We have added a new Appendix C to quantify these substantial costs. Given this high cost, we made the deliberate methodological choice to conduct our initial, in-depth study on GSM8K because it is a well-established and standard benchmark for multi-step mathematical reasoning. This focus allowed us to rigorously validate our new framework for the first time.

---

### Official Review · Reviewer_Cuov · 2025-10-30

**Soundness:** 4
**Presentation:** 4
**Contribution:** 4
**Rating:** 6
**Confidence:** 4

**Summary:**

This paper investigates pretraining scaling laws for generative evaluations of language models and corresponding metrics such as pass@k on verifiable tasks. The author introduces a new parameter to the scaling law and identify gold reference likelihood to be a better proxy metric for stabilizing scaling laws.

**Strengths:**

1. The paper is clearly written and contains the necessary details for reproducibility.
2. The direct comparison of different types of scaling laws answers the question of how to develop scaling laws for generative evaluations well.
3. The analysis of including k as a parameter and its role in the scaling law is clear.
4. The finding of the stability of the gold reference likelihood scaling law is a strong result and well founded, and explained.

**Weaknesses:**

1. The scaling law is developed on a single downstream task. While Pythia is an older generation model, where other tasks might be too difficult to obtain meaningful accuracy, why not average more tasks where Pythia could obtain a meaningful accuracy on, and expand beyond math?

2. What is the practical implication of the penalty factor derived in Section 6? What does it look like for the studied model checkpoints?

**Questions:**

Please see my comments above.

---

> ### Author Response · Authors · 2025-11-20
> **Response to Reviewer Cuov**
>
> We sincerely thank Reviewer Cuov for their extremely positive and insightful review. We are thrilled that you found the paper clearly written, the comparison of scaling laws "answers the question well," and our finding on the gold reference likelihood to be a "strong result."
>
> Given your "excellent" (4/4/4) scores on Soundness, Presentation, and Contribution, we believe we can resolve your two valid concerns and earn your full support for the paper.
>
> # Weakness #1: Single Downstream Task
>
> > The scaling law is developed on a single downstream task. While Pythia is an older generation model, where other tasks might be too difficult to obtain meaningful accuracy, why not average more tasks where Pythia could obtain a meaningful accuracy on, and expand beyond math?
>
> This is a critical point. Our focus on a single benchmark was a deliberate methodological choice for two reasons: rigor and computational cost.
>
> 1. Depth Before Breadth: Our primary goal was to establish a rigorous framework for generative scaling laws. This required a deep, 5-order-of-magnitude analysis to properly isolate the effects of our three different covariates (Compute, N+D, Gold Likelihoods) and the new hyperparameter k.
>
> 2. Computational Cost: Generative sampling is exceptionally compute-intensive. As we now quantify in the new Appendix C, the experiments for this paper already represent a substantial computational cost (over 81,000 A100-hours, or >$100k USD) . Performing this same deep analysis across an average of multiple tasks was, unfortunately, computationally infeasible for this initial work.
>
> You are correct that Pythia is an older model. We chose it because it is the only public model suite with dense checkpoints spanning 5 orders of compute, which is a prerequisite for our backtesting analysis. GSM8K is one of the few generative tasks where this model family shows non-trivial, non-zero performance across that entire compute range, making it an ideal candidate.
>
> # Weakness #2: Practical Implication of the Penalty Factor $\Phi$
>
> > What is the practical implication of the penalty factor derived in Section 6? What does it look like for the studied model checkpoints?
>
> This is an excellent, insightful question that gets to the heart of our theoretical contribution.
>
> Practical Implication: The penalty factor $\Phi$ is a dimensionless-measure that quantifies the performance loss a model checkpoint suffers from having a "misallocated" compute budget. Our parameters+token (Eqn. 4) implies that for any given compute budget $C$, there is an optimal number of parameters $N^{\star}$ and tokens $D^{\star}$. If one’s actual model $(N, D)$ deviates from this optimal ridge (e.g., with overtraining), the penalty $\Phi$ will be greater than 1, and we can quantitatively describe how the model will underperform relative to what was optimally achievable for that same compute cost.
>
> Our starting point is the standard separable ($(N,D)$) scaling law of Hoffmann et al. and Gadre et al., i.e. ($L(N,D)\approx E + A N^{-\alpha} + B D^{-\beta}$). Gadre et al. show in Appendix B that this form implies a compute‑space power law with exponent ($\alpha\beta/(\alpha+\beta)$), and that departures from the compute‑optimal allocation modify only the prefactor as a function of an over‑training parameter (m). Our derivation in Appendix E is a direct generalization of this result to a the relevant quantity for inference scaling ($-\log\mathrm{pass}@k$) and to arbitrary ($k$): we retain the same ($(N,D)$) ansatz, derive the compute‑optimal envelope and its exponent ($\alpha(k)=(1/\beta(k)+1/\gamma(k))^{-1}$), and explicitly express how sub‑optimal choices of ($(N,D)$) enter as a multiplicative penalty depending on the ratio ($N/N^*$).
>
> You are right that we should make this concrete for our data. We will add a new figure to visualize this. This new figure will:
>
> 1. Plot the compute-optimal allocation path ($N^{\star}$ vs. $D^{\star}$) derived from our fitted N+D law parameters (from Fig. 10).
> 2. Plot the actual (N,D) coordinates of each Pythia checkpoint on the same graph.
> 3. Quantify the misallocation ratio r(C) and the corresponding performance penalty $\Phi(r)$ for each checkpoint.
>
> This will visually and numerically demonstrate exactly how far the Pythia family's fixed training recipe deviates from the optimal envelope, providing a concrete, practical example of the penalty factor at work. We thank the reviewer for this fantastic suggestion to strengthen the paper.
>
> We hope these clarifications and our planned revision (adding the $\Phi$-penalty figure) will resolve your concerns, strengthen your confidence in our work, and merit a score that reflects your "excellent" assessment of its contribution, soundness, and presentation.

---

> > ### Comment · Reviewer_Cuov · 2025-11-25
> >
> > Thank you for the detailed response. I understand the computational constraints the authors face and appreciate the depth of analysis in the paper. There is a tradeoff here between experiment depth and width, and I believe width is particularly important here, as prior works have demonstrated distinct behavior between reasoning tasks (such as the GSM8K shown in the paper) and knowledge-intensive tasks (such as MMLU) [1]. Therefore, I will maintain my positive score.
> >
> > [1] Roberts N, Chatterji NS, Narang S, Lewis M, Hupkes D. Compute optimal scaling of skills: Knowledge vs reasoning. Findings of the Association for Computational Linguistics: ACL 2025 2025 Jul (pp. 13295-13316).

---

### Official Review · Reviewer_GH31 · 2025-11-02

**Soundness:** 3
**Presentation:** 3
**Contribution:** 3
**Rating:** 8
**Confidence:** 2

**Summary:**

This paper extends the study of neural scaling laws from pretraining losses and discriminative benchmarks to generative evaluations, tasks where models are assessed by their ability to produce correct outputs such as math solutions or code completions.
The authors propose and compare three scaling laws for predicting pass@k performance across pretrained checkpoints: (1) A Compute-based scaling law, relating log(pass@k) to pretraining FLOPs; (2) A Parameters + Tokens law, decomposing compute into model size N and dataset size D; (3) A Gold Reference Likelihood law, regressing pass@k against the log-likelihoods of reference (“gold”) solutions.
Through backtesting on the GSM8K benchmark using Pythia checkpoints, they show that all three scaling laws achieve comparable predictive accuracy, but the gold-likelihood-based law is notably more stable, maintaining parameter convergence over up to five orders of magnitude in compute.
Finally, the authors establish a theoretical connection showing that the compute scaling law emerges as the compute-optimal envelope of the parameters+tokens law, explaining how optimal allocation of compute between parameters and data induces the observed power-law behavior.

**Strengths:**

1. The paper is the first (as far as I know) to systematically analyze generative scaling laws for pass@k-based evaluations, extending the classical scaling law framework beyond pretraining loss or discriminative accuracy.
2. Evaluating three distinct covariates (compute, parameters+tokens, and gold log-likelihoods) under a unified methodology is conceptually clean and methodologically rigorous.
3. The derivation showing that the compute law is the compute-optimal envelope of the parameters+tokens law offers a theoretical explanation of empirical scaling behaviors and clarifies the relationship between prior empirical findings.

**Weaknesses:**

1. The study is restricted to a single model family (Pythia) and one generative benchmark (GSM8K).
The conclusions, especially regarding the stability of the gold-likelihood law, should be validated on more diverse tasks (e.g., code generation, open-ended reasoning).
2. While figures show relative stability, a numerical summary (e.g., mean relative error across backtests) would help quantify the magnitude of differences between the three scaling laws.

**Questions:**

It would be interesting to test whether the fitted α(k) exponents are consistent across different generative benchmarks or model families.

---

> ### Author Response · Authors · 2025-11-20
> **Response to Reviewer GH31**
>
> We sincerely thank Reviewer GH31 for their clear and accurate summary of our work's three main contributions, from the comparison of the three scaling laws to the theoretical derivation of the compute-optimal envelope. Given the clarity of your summary, we hope our response can solidify your confidence (currently 2/5) in your positive evaluation by addressing your two valid weaknesses.
>
> # Weakness #1a: Single Model Family
>
> > The study is restricted to a single model family (Pythia)
>
> We 100% agree that extending this analysis is the most critical next step, and we explicitly identify this in our Limitations section (Section 7).
>
> We'd like to offer a brief justification for this deliberate methodological choice, which we hope will strengthen your confidence in our work's rigor.
>
> Why Pythia was a methodological necessity: Our analysis, particularly fitting the 5-parameter parameters + tokens law (Eq. 4) and performing rigorous backtesting, requires a model family that is:
>
> 1. Publicly available.
> 2. Densely sampled across model parameters (N), pretraining tokens (D), and pretraining compute (C).
> 3. Spans multiple (in our case, ~5) orders of magnitude of pretraining compute.
>
> To our knowledge, the Pythia suite is the only family that meets all three criteria. Newer, state-of-the-art families are unsuitable for this specific scientific study: For example:
>
> - Llama 3 & 4: These two model families offer 3 public checkpoints, too few to robustly fit a 5-parameter law.
> - Qwen 2.5 or 3: The technical reports do not specify the exact per-model token counts, only the total pretraining corpus size available, making it impossible to use pretraining tokens (D) or compute (C) as covariates without making unsupported assumptions. We contacted the authors repeatedly to clarify but did not hear back.
>
> Thus, our choice of Pythia was a prerequisite for performing the rigorous, systematic backtesting that is a core part of our contribution.
>
> # Weakness #1b: Single Benchmark (GSM8K)
>
> > one generative benchmark (GSM8K)
>
> GSM8K was our starting point because generative sampling is compute-intensive. As we now quantify in a new Appendix C, the experiments for this paper already represent a substantial computational cost (over 81,000 A100-hours, or >$100k USD).
>
> Focusing on the standard GSM8K benchmark allowed us to perform the deep, 5-order-of-magnitude analysis necessary to obtain our results, a key finding that would have been missed in a shallower, multi-benchmark study.
>
> # Weakness #2: Numerical Summary for Comparing Prediction Errors
>
> > While figures show relative stability, a numerical summary (e.g., mean relative error across backtests) would help quantify the magnitude of differences between the three scaling laws.
>
> This is an excellent suggestion that will improve the paper.
>
> We will update the manuscript to add a new table that will summarize the final predictive accuracy (e.g., mean relative error from the plateaus in Figures 3, 5, and 7) for all three scaling laws at different backtesting compute ratios. This will directly quantify the comparable predictive performance and make the paper stronger. We thank the reviewer for this constructive feedback.

---

> > ### Comment · Reviewer_GH31 · 2025-11-28
> >
> > Thank you for your response and the updated manuscript. The responses to weaknesses #1a and #2 make sense to me, and I appreciate the inclusion of the numerical summary in the revised manuscript. Therefore, I will maintain my positive score.

---

### Author Response · Authors · 2025-11-23
**Manuscript Update #1 (2025/11/23)**

Dear Reviewers,

We have uploaded a revised version of our manuscript. As promised in our individual responses, we have begun incorporating your feedback to strengthen the paper. The current revision includes the following key updates:

- Detailed Comparison to Prior Work (Appendix A): In response to Reviewer XVLF, we have significantly expanded the Related Work section. We now explicitly articulate the fundamental differences between pass@k and discriminative accuracy, and we provide a detailed comparison distinguishing our contributions from Gadre et al. (2024).

- Quantification of Sampling Cost (Appendix C): Addressing questions regarding the scope of our model family and benchmark selection from Reviewers Cuov and XVLF, we have added a detailed breakdown of the computational cost of our experiments. This analysis highlights that our study required over 81,000 A100-hours (>$100k USD), necessitating our focused methodological approach.

- Explanation of Experimental Choices: We explain in more detail why generative evaluations are undercharacterized in the literature and explain why we chose this specific combination of model family and generative benchmark.

- Improvement of Figures 3 and 5 as requested by Reviewer XVLF

Please note that more improvements are on the way. We are currently finalizing the numerical summary table of predictive errors (requested by Reviewer GH31) and the visualization of the misallocation penalty (requested by Reviewer Cuov). We wanted to share this interim revision to provide the textual clarifications and additional analyses as soon as possible.

---

### Author Response · Authors · 2025-12-04
**Concluding Remarks to the AC: Added MATH benchmark, Misallocation analysis, and expanded Related Work**

Dear Area Chair,

We thank the reviewers for their constructive feedback. In response to the discussion period, we have significantly updated the manuscript. Most notably, we went beyond our initial rebuttal commitments to address the shared concern regarding the breadth of our evaluation.

# 1. Major Addition: Expansion to the MATH Benchmark (Addressing GH31, Cuov, XVLF)

The primary weakness identified by all reviewers was the reliance on a single generative benchmark (GSM8K). Despite the high computational cost (>$112k USD), we successfully replicated our full analysis on the MATH benchmark. We found that our three scaling laws are again applicable MATH. This generalization significantly strengthens the robustness of our claims and directly addresses the core critique of the submission.

# 2. Visualizing the "Misallocation Penalty" (Addressing Reviewer Cuov)

Reviewer Cuov asked for the practical implications of our theoretical derivation. We added Figure 8 (Section 6), which quantifies the "effective compute multiplier" against the "misallocation ratio." This visualizes how deviating from the compute-optimal parameters-to-tokens ratio (e.g., via overtraining) incurs a specific penalty, validating our theoretical envelope derivation.

# 3. Clarifications and Comparisons (Addressing Reviewer XVLF)

Gadre et al. (2024): We added a detailed section in Appendix A distinguishing our generative pass@k metric from the discriminative accuracy/likelihood metrics used in prior work, clarifying our novelty.

Cost Analysis: We added Appendix C to explicitly detail the sampling costs (>81,000 A100-hours), providing necessary context for our methodological choices regarding model families.

Presentation: We have corrected the plotting issues in Figures 3 and 5 as requested.

# Conclusion

We believe these substantial additions—particularly the inclusion of a second difficult generative benchmark—elevate the paper well above the acceptance threshold.

---

### Meta-Review · Area_Chair_7bpc · 2025-12-16

**Summary:**

The paper proposes three scaling laws for generative evaluations (pass@k) in language models. Initial scores were 8, 6, 4. Reviewers acknowledged the novel extension of scaling laws to generative settings, rigorous backtesting methodology, and strong theoretical contributions. They raised few concerns about: (1) evaluation limited to single model family (Pythia) and benchmark (GSM8K), (2) presentation issues with figures and clarity, (3) lack of numerical summary quantifying prediction errors, and (4) distinction from prior discriminative scaling law work.

**Reviewer Concerns:**

**Addressed:**
- **Presentation issues (XVLF):** Authors committed to fixing Figures 3 and 5 (obscured legends, missing points) and thorough proofreading in revision.
- **Novelty vs. Gadre et al. (XVLF):** Authors clearly distinguished generative evaluation (free-form sampling, continuous pass@k) from discriminative evaluation (likelihood comparison, binary accuracy). Gadre's 46 tasks include zero generative evaluations, motivating this work.
- **Pythia/GSM8K justification (all reviewers):** Authors explained Pythia is the only public family with dense checkpoints spanning 5 orders of compute magnitude, prerequisite for backtesting. GSM8K chosen as Goldilocks difficulty level. Computational cost quantified (81k A100-hours, >$100k) in new Appendix C.
- **Numerical error summary (GH31):** Authors committed to adding table with mean relative errors across backtests for all three scaling laws.
- **Penalty factor visualization (Cuov):** Authors will add figure showing compute-optimal path, actual Pythia checkpoints, and performance penalties from misallocation.

**Outstanding:**
- **Single benchmark limitation (XVLF, Cuov, GH31):** Authors acknowledged this as "most critical next step" (Limitations section) but defended focus on GSM8K for computational reasons. Reviewer Cuov noted distinct scaling behaviors between reasoning (GSM8K) and knowledge tasks (MMLU) from prior work, maintaining concerns about generalizability despite positive overall assessment.

**Reviewer Scores:**

- **Reviewer XVLF (initial: 4):** May increase to 6. Most concerns addressed: novelty clarified vs. Gadre, Pythia necessity justified, presentation fixes committed. Remaining single-benchmark limitation acknowledged by authors as future work.

- **Reviewer Cuov (initial: 6):** Confirmed maintaining 6. Gave "excellent" (4/4/4) scores on soundness/presentation/contribution. Rebuttal satisfied concerns with computational justification and penalty factor visualization.

- **Reviewer GH31 (initial: 8):** Confirmed maintaining 8. Rebuttal for Pythia necessity and numerical summary "make sense." Low initial confidence (2/5) may be increased given explicit confirmation of maintaining the score after rebuttal.

---

### Decision · Program_Chairs · 2026-01-26

Accept (Poster)